# Effects of neuromuscular gait modification strategies on indicators of knee joint load in people with medial knee osteoarthritis: A systematic review and meta-analysis

M. Denika C. Silva[1,2,3]*, Diana M. Perriman[1,2,4], Angela M. Fearon[1,2,5], Daniel Tait[1,5], Trevor J. Spencer[1,5], Dianne Walton-Sonda[6], Milena Simic[7], Rana S. Hinman[8], Kim L. Bennell[8], Jennie M. Scarvell[1,2]

1 Faculty of Health, University of Canberra, Bruce, Australia, 2 Trauma and Orthopaedic Research Unit, Canberra Hospital, Canberra, Australia, 3 Department of Physiotherapy, General Sir John Kotelawala Defence University, Werahera, Sri Lanka, 4 College of Medicine and Health Sciences, Australian National University, Canberra, Australia, 5 Research Institute for Sport and Exercise, University of Canberra, Bruce, Australia, 6 Library and Multimedia Services, Canberra Hospital, Canberra, Australia, 7 Faculty of Medicine and Health, The University of Sydney, Sydney, Australia, 8 Centre for Health, Exercise & Sports Medicine, Department of Physiotherapy, School of Health Sciences, University of Melbourne, Melbourne, Australia

* denika.silva@canberra.edu.au

## Abstract

### Objectives

This systematic review aimed to determine the effects of neuromuscular gait modification strategies on indicators of medial knee joint load in people with medial knee osteoarthritis.

### Methods

Databases (Embase, MEDLINE, Cochrane Central, CINAHL and PubMed) were searched for studies of gait interventions aimed at reducing medial knee joint load indicators for adults with medial knee osteoarthritis. Studies evaluating gait aids or orthoses were excluded. Hedges' g effect sizes (ES) before and after gait retraining were estimated for inclusion in quality-adjusted meta-analysis models. Certainty of evidence was assessed using the Grading of Recommendations Assessment, Development and Evaluation (GRADE) approach.

### Results

Seventeen studies (k = 17; n = 362) included two randomised placebo-controlled trials (RCT), four randomised cross-over trials, two case studies and nine cohort studies. The studies consisted of gait strategies of ipsilateral trunk lean (k = 4, n = 73), toe-out (k = 6, n = 104), toe-in (k = 5, n = 89), medial knee thrust (k = 3, n = 61), medial weight transfer at the foot (k = 1, n = 10), wider steps (k = 1, n = 15) and external knee adduction moment (KAM) biofeedback (k = 3, n = 84). Meta-analyses found that ipsilateral trunk lean reduced early stance peak KAM (KAM1, ES and 95%CI: -0.67, -1.01 to -0.33) with a dose-response effect and reduced KAM impulse (-0.37, -0.70 to -0.04) immediately after single-session training.

**Data Availability Statement:** All the data are presented within the oaper and its Supporting information files.

**Funding:** The authors received no specific funding for this work.

**Competing interests:** There are no further conflicts of interest to disclose. All the competing interests are included. This does not alter our adherence to all PLOS ONE policies on sharing data and materials.

Toe-out had no effect on KAM1 but reduced late stance peak KAM (KAM2; -0.42, -0.73 to -0.11) immediately post-training for single-session, 10 or 16-week interventions. Toe-in reduced KAM1 (-0.51, -0.81 to -0.20) and increased KAM2 (0.44, 0.04 to 0.85) immediately post-training for single-session to 6-week interventions. Visual, verbal and haptic feedback was used to train gait strategies. Certainty of evidence was very-low to low according to the GRADE approach.

## Conclusion

Very-low to low certainty of evidence suggests that there is a potential that ipsilateral trunk lean, toe-out, and toe-in to be clinically helpful to reduce indicators of medial knee joint load. There is yet little evidence for interventions over several weeks.

## 1. Introduction

Non-surgical management strategies for knee osteoarthritis (OA) have become a high priority with increasing prevalence [1, 2]. Besides, knee OA commonly occurs in the medial compartment of the joint [3]. As new strategies and programs emerge, a comprehensive understanding of which non-surgical strategies have the potential for arresting or slowing knee OA progression is urgently required [2]. Gait retraining may have the potential to slow disease progression by reducing knee joint load since knee joint load is associated with the progression of medial knee OA [4, 5].

Increased knee joint load on the medial compartment of the knee is associated with the progression of medial knee OA [5]. As direct measurement of knee joint load is impractical given that it would require an invasive procedure, surrogate measures are typically adopted. The external knee adduction moment (KAM), evaluated using 3-dimensional gait analysis, is frequently used as a surrogate measure of medial knee joint load [6]. Early stance peak KAM is reported to predict 63% of medial knee joint load in the stance phase of gait [7]. Other biomechanical parameters that have been shown to contribute significantly to the medial knee joint load include the early stance peak knee flexion moment (KFM1) [7], the late stance peak KAM (KAM2), the KAM impulse (integration of the KAM over stance time), and the late stance peak knee flexion moment (KFM2) [8]. Since higher medial knee joint load is associated with knee OA progression [5, 9], this is an appropriate target for interventions.

Gait modification strategies to reduce knee joint load include gait aids, orthoses and neuromuscular gait modification strategies [10]. Some of the neuromuscular gait modification strategies that have been studied include: increased lateral trunk flexion towards the symptomatic knee during the stance phase of gait (ipsilateral trunk lean), increased and decreased foot progression angle (FPA, toe-out and toe-in respectively), medialising the knee during the stance by a combination of hip internal rotation and adduction (medial knee thrust), and increasing the lateral distance between the feet (increased step width) [11, 12]. These strategies have demonstrated some ability to reduce the indicators of medial knee joint load such as KAM. However, a comprehensive synthesis of gait modifications would assist their implementation in clinical practice.

Three previous systematic reviews have analysed the efficacy of gait modification strategies on medial knee joint load [13–15]. In 2011, Simic et al. reviewed 24 gait retraining studies, fourteen of which investigated healthy participants without knee OA [13]. A review by Bowd et al. (2019) specifically investigated whether gait modifications aimed at negative

consequences for loads at the hip and ankle [14]. Wang et al. (2020) investigated effects of toe-out and toe-in strategies [15] but did not explore any of the other common strategies such as trunk lean, medial knee thrust etc. Recently, the field has advanced to incorporate innovative feedback strategies such as haptic sensors [16] and real-time feedback on knee joint moments [17]. Our systematic review adds to previous reviews by including all neuromuscular gait modification strategies and exclusively in people with medial knee OA.

Therefore, the aim of this systematic review was to determine the effects of neuromuscular gait modification strategies on indicators of medial knee joint load in people with medial knee OA.

## 2. Materials and methods

This systematic review was conducted according to the Preferred Reporting Items for Systematic Reviews and Meta-Analyses (PRISMA) guidelines [18] and registered in PROSPERO (registration number: CRD42020153962).

### 2.1. Literature search

Databases (Embase, MEDLINE, CENTRAL (Cochrane Central Register of Controlled Trials), CINAHL and PubMed) were searched from their inception to March 2021. The search strategy was as follows: ((*knee OR genu OR tibiofemoral) AND (osteoarthr* OR degenerative)) AND (gait* OR walk* OR ambulat* OR locomot*) AND (train* OR retrain* OR educat* OR reeducat* OR intervent* OR modif* OR strateg* OR pattern* OR rehab*) AND (biomechanic* OR kinematic * OR (knee* adduct* moment*) OR KAM OR varus thrust* OR load* OR force* OR moment*). Further, database-specific MeSH terms were used (S1 Appendix). An additional manual search was performed of the reference lists of included studies.

### 2.2. Study selection

Studies of any design that included participants with medial compartment knee OA (confirmed by imaging), who were taught a new walking pattern and included pre and post-intervention measurements of medial knee joint load indicators, were included (Table 1). Studies examining the effects of gait aids or orthoses were excluded. Covidence software (Veritas

**Table 1. Criteria for the eligibility of papers included in the systematic review.**

| Inclusion criteria | Exclusion criteria |
|---|---|
| 1. Any study design (e.g. randomised controlled trials, quasi clinical trials, cohort studies, case series, studies with or without a control group) | 1. No original data (e.g. a review or editorial) |
| 2. Adults aged 18 years or older | 2. Abstracts only and other materials not published as a full peer-reviewed paper |
| 3. Medial compartment knee osteoarthritis confirmed by imaging | 3. Predominantly lateral compartment knee osteoarthritis |
| 4. Any intervention where the participants are taught a new walking pattern that is aimed at reducing the load on the medial compartment of the knee and its effects can be determined in isolation from other intervention effects. | 4. Predominantly patellofemoral knee osteoarthritis |
| 5. Within-subject measures of gait before and after intervention were recorded | 5. Concurrent osteoarthritis in other lower limb joints unless data are reported separately |
| 6. Outcomes were indicators of medial knee joint load | 6 Interventions with gait aids or orthoses |
| | 7. Intervention effects cannot be determined in isolation from other intervention effects |

Health Innovation, Melbourne, Australia (www.covidence.org) was used to manage the review process. Articles identified in the search were uploaded and duplicates were removed. Titles and abstracts of studies, followed by full text, were screened independently by two reviewers and any conflicts were resolved by consulting with a third reviewer.

## 2.3. Methodological quality appraisal

The risk of bias was assessed independently by two reviewers using the Downs and Black checklist [19] and differences resolved by a third reviewer. This checklist has 27 items across 5 subscales: reporting, external validity, internal validity-bias, internal validity-confounding (cohort selection bias), and power. The 'power' subscale (Question 27) was removed from the quality assessment due to item ambiguity [20]. We graded the quality of each paper in terms of total points scored (poor: ≤14, fair: 15 to 19, good: 20 to 24, excellent: ≥25) [21]. The quality effects score (Qi) (total points divided by the maximum points) of each study was calculated and used in meta-analyses to adjust for quality in all models.

## 2.4. Data extraction and synthesis

Data extracted included: design, participants, details of interventions, outcomes and times of assessment. The primary outcomes were indicators of medial knee joint load during gait measured via 3-dimensional gait analysis. We extracted KAM1, KAM impulse, [16, 22], KAM2, KFM1 and KFM2 [7]. Secondary outcomes were 3D knee kinematic data measuring flexion-extension, abduction-adduction, and internal-external rotation angles. We extracted outcomes as mean and standard deviation (SD). When studies reported alternative measures such as standard error (SE) or confidence intervals (CI), we calculated SD using validated statistical methods [23].

Meta-analyses were conducted to determine the effects of gait modification strategies on indicators of medial knee joint load using MetaXL software (version 5.3-EpiGear Wilston, Queensland, Australia). For this review, we refer to 'gait strategies' as the umbrella term describing the gait modification (e.g. trunk lean, toe-in, toe-out). Within the gait strategy implemented, different doses (or degrees) were implemented in studies. Meta-analyses were performed where there was a minimum of three studies using similar gait modification strategies. Single case studies were not included in the meta-analyses because of the potential for bias and effect size cannot be calculated. We used the mean, SD and sample size for pre-and post-test data to calculate Hedges' g effect sizes (ES) with 95% CI. A quality effects model based on the inverse variance fixed-effect model was used for the main analysis. A fixed-effect model was employed because, when using the quality effects method, it outperforms the random effects estimator and avoids underestimation of statistical error [24]. In this model, the redistribution of inverse variance weights is achieved using a quality parameter between zero (lowest-quality) and one (highest-quality) [25].

Where a single study reported multiple doses of the same gait strategy, a single representative dose that was most commonly used by included studies was selected for the meta-analysis (for example, if a study implemented different doses of toe-out, we selected a single representative dose that was most commonly used by included studies, and the overall effect was calculated using that data).

The overall effect was considered significant if the 95%CI did not cross the line of null effect (zero) in the forest plot. The results were interpreted in terms of the effect size (small: 0.2 to 0.49, medium: 0.5 to 0.79 and large: ≥ 0.8) [26]. Statistical heterogeneity ($I^2$) values were estimated but may be biased if a smaller number of studies was included in meta-analyses [27].

Publication bias was not assessed using funnel plots or Egger's regression test because we did not find the minimum requirement of 10 papers [23].

## 2.5. Certainty of evidence

We used the Grading of Recommendations Assessment, Development and Evaluation (GRADE) approach to assess the certainty of the evidence for each gait strategy. The GRADE approach considers the risk of bias, inconsistency/heterogeneity, indirectness, imprecision of the evidence and publication bias to downgrade the certainty, while evidence of a large effect, dose-response, and the effect of plausible residual confounding is used to upgrade the certainty [28, 29]. Where there were several study designs, we considered the certainty of evidence is low due to methodological heterogeneity. The evidence was categorised as very-low, low, moderate or high certainty based on the above criteria.

## 3. Results

The search yielded 2081 records after the removal of duplicates and 17 studies were eligible (k = 17) after screening (Fig 1). No additional studies were identified in the manual search of reference lists of included studies.

### 3.1. Quality assessment

The overall quality of studies was good to fair (total point scored, mode = 19, range 13 to 24) except for two case studies [30, 31] (Table 2). The studies performed generally well in subscales of reporting, internal validity-bias and internal validity-confounding (cohort selection bias). The studies performed poorly in the external validity subscale (Table 2).

### 3.2. Study characteristics and details of gait strategies

The 17 studies (total sample size (n) of 362) included two randomised placebo-controlled trials [11, 32], four within-session, randomised cross-over trials [12, 33–35], two case studies [30, 31] and nine pre-post-test cohort studies [16, 17, 36–42]. The studies consisted of gait strategies of ipsilateral trunk lean (k = 4, n = 73), toe-out (k = 6, n = 104), toe-in (k = 5, n = 89), medial knee thrust (k = 3, n = 61), medial weight transfer at the foot (k = 1, n = 10), wider steps (k = 1, n = 30), self-directed gait or combination of strategies (example, toe-in plus increased step width) with specific KAM biofeedback (k = 3, n = 84). Studies varied how they implemented these strategies, for example, different doses of toe-out or medial knee thrust with or without feedback. Study characteristics are shown in Table 3.

A variety of feedback methods were used in included studies, for example, ink-lines on the floor [37], verbal feedback [33], active haptic feedback [16] and real-time visual feedback [12]. Five studies used visual or audio feedback to train strategies to reduce KAM1 to a target value (10% or 20%) (specific KAM feedback) [17, 32, 36, 40, 42] (Table 3).

The duration of interventions ranged from single-session to nine months (Table 3). The timing of the outcome assessment varied from immediately following the training of the intervention to six months follow-up periods (after completion of the program) (Table 3).

### 3.3. Effects on knee joint load

The indicators of medial knee joint load reported in the studies were KAM1 (k = 17), KAM impulse (k = 9), KAM2 (k = 11), KFM1 (k = 11), and KFM2 (k = 2) (Table 4).

   **3.3.1. Meta-analyses.**   Fourteen studies were included in the meta-analyses [11, 12, 16, 17, 33–42] reporting ipsilateral trunk lean, toe-out and toe-in gait strategies. Three studies were

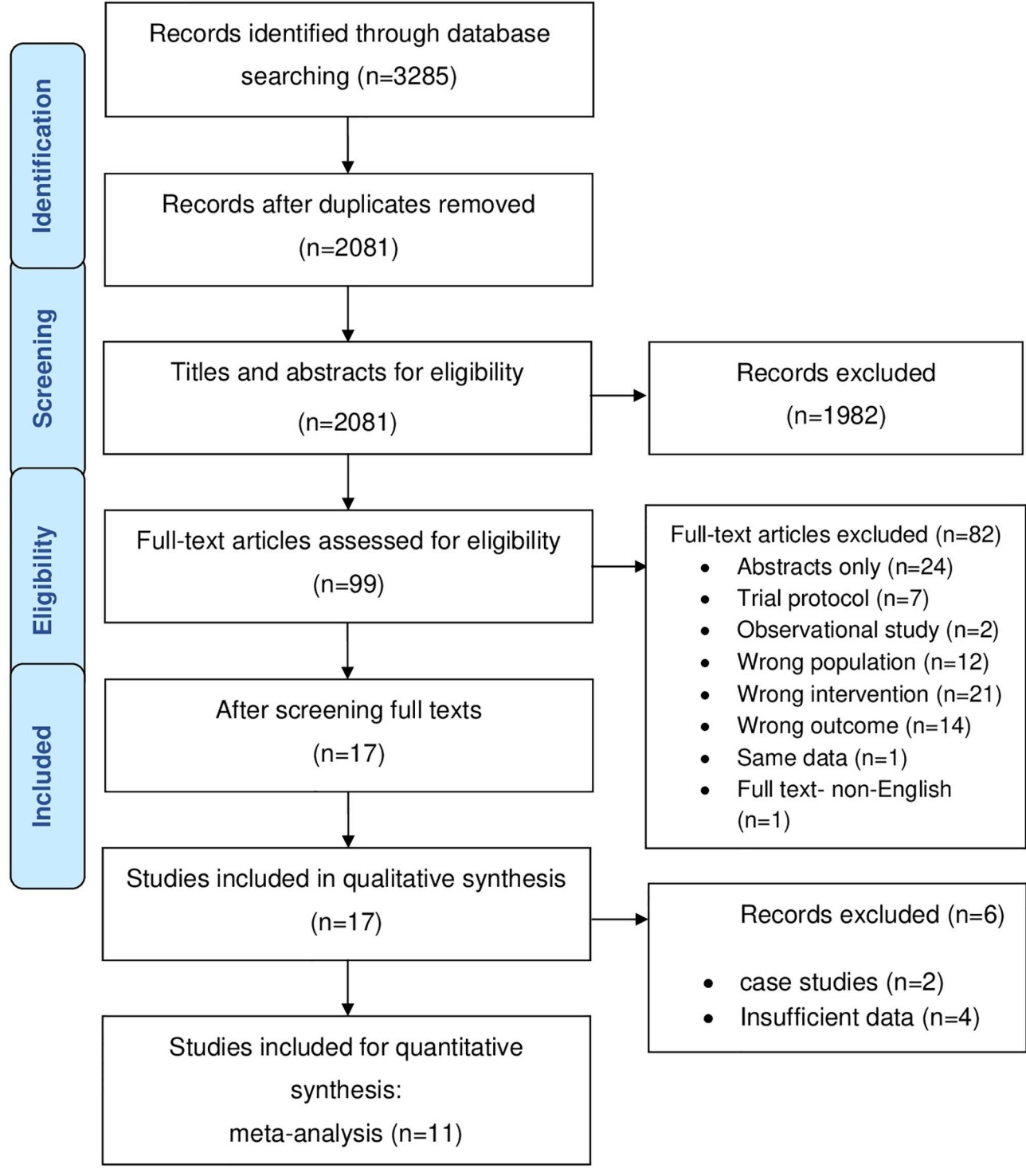

**Fig 1. PRISMA flow diagram of study selection.**

**Table 2. Methodological quality of included studies assessed using Downs and Black scale [19].**

| Study (Author, year) | Reporting (out of 11) | External validity (out of 3) | Internal validity- bias (out of 7) | Internal validity- Confounding (out of 6) | Total score (out of 27) | Quality effect score (Qi) |
|---|---|---|---|---|---|---|
| **Booij et al, 2020** [36] | 10 | 0 | 5 | 4 | 19 | 0.70 |
| **Charlton et al, 2019** [35] | 10 | 0 | 5 | 4 | 19 | 0.70 |
| **Cheung et al, 2018** [32] | 11 | 1 | 6 | 5 | 23 | 0.85 |
| **Erhart-Hledik et al, 2017** [16] | 9 | 0 | 5 | 4 | 18 | 0.67 |
| **Fregly et al, 2007** [30] | 6 | 0 | 4 | 3 | 13 | 0.48 |
| **Gerbrands et al, 2017** [33] | 9 | 0 | 5 | 4 | 18 | 0.67 |
| **Guo et al, 2006** [37] | 8 | 0 | 5 | 3 | 16 | 0.59 |
| **Hunt et al, 2011** [31] | 7 | 0 | 4 | 3 | 14 | 0.52 |
| **Hunt and Takacs, 2014** [38] | 11 | 1 | 5 | 3 | 20 | 0.74 |
| **Hunt et al, 2018** [11] | 10 | 2 | 6 | 6 | 24 | 0.89 |
| **Richards et al, 2018 a** [42] | 8 | 0 | 5 | 3 | 16 | 0.59 |
| **Richards et al, 2018 c** [17] | 10 | 0 | 5 | 3 | 18 | 0.67 |
| **Shull et al, 2013 a** [39] | 10 | 0 | 5 | 3 | 18 | 0.67 |
| **Shull et al, 2013 b** [40] | 9 | 1 | 5 | 4 | 19 | 0.70 |
| **Simic et al, 2012** [12] | 9 | 0 | 5 | 3 | 17 | 0.63 |
| **Simic et al, 2013** [34] | 9 | 0 | 5 | 4 | 18 | 0.67 |
| **Tokuda et al, 2018** [41] | 10 | 0 | 5 | 4 | 19 | 0.70 |

The range of scores possible for each subscale are; Reporting: 0 to 11, External validity: 0 to 3, Internal validity- bias: 0 to 7, Internal validity- Confounding: 0 to 6, Total score: 0 to 27 and Quality effect score (Qi): 0 to 1.

excluded because the effect size could not be calculated (two case studies [30, 31] or reported only percentage differences as outcomes [32]).

Ipsilateral trunk lean reduced KAM1 and KAM impulse immediately after single-session training. The studies [12, 33, 41] implemented five different doses of ipsilateral trunk lean (6˚, 9˚, 10˚,12˚ and 'to the greatest possible extent') noting that three doses (6˚, 9˚, and 12˚) were implemented in the same participants [12]. From that study, we selected 9˚ trunk lean. Ipsilateral trunk lean reduced KAM1 with a medium overall effect (ES = -0.67, CI = -1.01, -0.33, k = 3, n = 72) and the KAM impulse was reduced with a small overall effect (ES = -0.37, CI = -0.70, -0.04, k = 3, n = 72) immediately after single-session training (Fig 2A and 2B). A dose-response effect was evident (larger lean angles produced greater reductions) for KAM1 but not for KAM impulse. Statistical heterogeneity values of $I^2 = 0$, may be biased as there were very few studies included (k = 3) [27]. There were insufficient studies for meta-analysis of ipsilateral trunk lean for KAM2, KFM1 and KFM2, however, individual studies showed reduced KAM2, KFM1 and KFM2 (Table 4).

Toe-out reduced KAM2 immediately after training, but not KAM1 or KAM impulse, with intervention duration lasting from single-session to 4-months. The studies [11, 34, 35, 37, 38] implemented four doses of toe-out (10˚, 15˚, 20˚ and 30˚) noting that one study [34] implemented three doses of toe-out in the same participants and another study [35] implemented two doses in the same participants (Table 3). From these studies, we selected 20˚ toe-out.

**Table 3. Details of gait modification programs of included studies.**

| Study, (Author, Year) | Design | Participants recruited (completed) | Participant details: KL grade: n (participants), Age, years, Mean (SD) BMI, kg/m², Mean (SD) | Gait modification strategy | Gait implementation and feedback | Duration | Assessment time point/s | Adverse effects |
|---|---|---|---|---|---|---|---|---|
| Booij et al, 2020 [36] | Pre-post-test | 30 (27) | I: 12, II: 7, III: 7, IV: 4, 62.7 (5.9), 25.5 (2.7) | Toe-in | A. Toe-in: individualised to reduce KAM1 by ≥ 10% (Specific KAM feedback by real-time, visual) | 1 session | Immediate | NR |
| | | 30 (22) | I: 12, II: 7, III: 7, IV: 4, 62.7 (5.9), 25.5 (2.7) | Wider steps | B. Wider steps: individualised to reduce KAM1 by ≥10% (Specific KAM feedback by real-time, visual) | 1 session | Immediate | NR |
| | | 30 (28) | I: 12, II: 7, III: 7, IV: 4, 62.7 (5.9), 25.5 (2.7) | Medial knee thrust | C. Medial knee thrust: individualised to reduce KAM1 by ≥ 10% (Specific KAM feedback by real-time, visual) | 1 session | Immediate | NR |
| Charlton et al, 2019 [35] | Randomised cross-over | 15 (13) | I: 0, II: 7, III: 8, IV: 0, 67.9 (9.4), NR, mean (SD) of weight 75.6 (15.0) kg and height 1.67 (0.11) m | Toe-in | A. Toe-in: +10 degrees (Real-time, visual) | 1 session | Immediate | NR |
| | | 15 (11) | | | B. FPA: 0 degrees (Real-time, visual) | 1 session | Immediate | NR |
| | | 15 (15) | | Toe-out | C. Toe-out: +10 degrees (Real-time, visual) | 1 session | Immediate | NR |
| | | 15 (14) | | | D. Toe-out: +20 degrees (Real-time, visual) | 1 session | Immediate | NR |
| Cheung et al, 2018* [32] | RCT-assessor blind | Total- 23 (Gait retraining- 12, Walking exercise- 11) (Total-20 (Gait retraining- 10, Walking exercise- 10)) | I: 2, II: 8, III: 0, IV: 0 (Gait retraining), I: 3, II: 7, III: 0, IV: 0 (Walking exercise); 60.8 (6.4) (Gait retraining), 63.1 (5.9) (Walking exercise); 24.5 (2.4) (Gait retraining), 25.2 (1.1) (Walking exercise) | Self-selected | Self-selected: Adjust either foot progression angle, hip adduction/rotation, and/or trunk sway with visual feedback: to reduce KAM1 by 20% (Specific KAM feedback by real-time, visual) | 6 weeks | Immediate, 6 months | None |
| Erhart-Hledik et al, 2017 [16] | Pre-post-test | 10 (10) | I< = All 65.3 (9.8) 27.8 (3.0) | Medial weight transfer at the foot | A. Medial weight transfer at the foot (Active, haptic) | 1 session | Immediate | NR |
| | | | | | B. Medial weight transfer at the foot: + increased gait speed (Active, haptic) | 1 session | Immediate | NR |
| Fregly et al., 2007# [30] | Case study | 1 (1) | I: 0, II: 1, III: 0, IV: 0, 37, 23.9 | Medial knee thrust | Medial knee thrust (Studying plots and animated results based on computer model data) | 9 months | Immediate | NR |

(*Continued*)

**Table 3.** (Continued)

| Study, (Author, Year) | Design | Participants recruited (completed) | Participant details: KL grade: n (participants), Age, years, Mean (SD) BMI, kg/m², Mean (SD) | Gait modification strategy | Gait implementation and feedback | Duration | Assessment time point/s | Adverse effects |
|---|---|---|---|---|---|---|---|---|
| **Gerbrands et al, 2017** [33] | Randomised cross-over | 30 (29) | NR, mean (SD) KOOS pain and function: 57.5 (13.4) and 62.3 (14.1), 61 (6.2), NR, mean (SD) of weight 75.7 (13.1) kg and height 1.71 (0.1) m | Trunk lean** | A. Trunk lean**: to the greatest possible extent (Clinician, verbal) | 1 session | Immediate | NR |
| | | | | Medial knee thrust | B. Medial knee thrust: to the greatest possible extent (Clinician, verbal) | 1 session | Immediate | NR |
| **Guo et al, 2006** [37] | Pre-post-test | 10 (9) | between I-III: All, 64 (8), 29.0 (5.6) | Toe out | Toe out: +15 degrees (Visual, ink line) | 1 session | Immediate | NR |
| **Hunt et al, 2011** [31] | Case study | 1 (1) | I: 0, II:0, III: 1, IV: 0, 64, 23.6 | Trunk Lean** | A. Trunk Lean**: self-selected angle (Clinician, verbal) | 1 session | Immediate | Some difficulty |
| | | | | Toe-out | B. Toe-out: self-selected angle (Clinician, verbal) | 1 session | Immediate | Moderate difficulty |
| **Hunt and Takacs, 2014[†]** [38] | Pre-post-test | 16 (15) | I: 0, II: 4, III: 9, IV: 3, 64.8 (10.4), 29.9 (6.8) | Toe-out | Toe-out: +10 degrees (Real-time, visual feedback) | 10 weeks | immediate | Joints discomfort (hip, knee, ankle) in first two weeks |
| **Hunt et al, 2018** [11] | RCT-assessor blind | Total- 79 (Gait retraining- 40, Progressive walking- 39) (Total- 67 (Gait retraining- 35, Progressive walking- 32)) | I: 0, II: 19, III: 17, IV: 4 (Gait retraining), I: 0, II: 18, III: 14, IV: 7 (Progressive walking); 64.6 (7.6) (Gait retraining), 65.4 (9.6) (Progressive walking); 27.3 (3.5) (Gait retraining), 27.4 (3.5) (Progressive walking) | Toe-out | Toe-out: +15 degrees (Mirror guided biofeedback) | 4 months | Immediate, 1 month | Hip pain (in 3–8 weeks), big toe pain (in the intervention), posterior thigh pain (in 3 weeks) |
| **Richards et al, 2018 a*** [42] | Pre-post-test | 40 (40) | I: 19, II: 8, III: 9, IV: 4, 61.7 (6.0), 25.6 (2.5) | Self-selected gait modification then, combination of gait strategies | A. Self-selected: to reduce KAM1 by 10% (Specific KAM feedback by real-time, visual) | 1 session | Immediate | NR |
| | | | | | B. Self-selected: to reduce KAM1 by 10% (Specific KAM feedback by real-time, audio) | 1 session | Immediate | NR |
| | | | | | C. Combination of gait strategies (Toe-in, increased step- width and medial knee thrust with KAM feedback: to reduce KAM1 by 10% (Specific KAM feedback by real-time, visual) | 1 session | Immediate | NR |

(*Continued*)

**Table 3.** (Continued)

| Study, (Author, Year) | Design | Participants recruited (completed) | Participant details: KL grade: n (participants), Age, years, Mean (SD) BMI, kg/m², Mean (SD) | Gait modification strategy | Gait implementation and feedback | Duration | Assessment time point/s | Adverse effects |
|---|---|---|---|---|---|---|---|---|
| **Richards et al, 2018 c*** ‡ [17] | Pre-post-test | 21 (16) | I: 14, II: 2, III: 4, IV: 1, 61.3 (5.73), 25.4 (2.6) | Combination of gait strategies | Combination of gait strategies (Toe-in: +10 degrees (all participants), Step- width: between 15–20 cm (5 out of 21 participants) with KAM feedback: to reduce KAM1 by 10% modifying gait (Specific KAM feedback by real-time, visual) | 6 weeks | Week 1, immediate, 3 and 6 months | Muscle soreness, and hip or back pain |
| **Shull et al, 2013 a** [39] | Pre-post-test | 12 (12) | I: 0, II: 4, III: 7, IV: 1, 59.8 (12.0), 26.5 (4.2) | Toe-in | Toe-in: +5 degrees (Real-time, vibration) | 1 session | Immediate | NR |
| **Shull et al, 2013 b** [40] | Pre-post-test | 10 (10) | I: 0, II: 3, III: 6, IV: 1, 60 (13), 26.6 (4.7) | Toe-in | Toe-in: to reduce KAM1 by 10% (Specific KAM feedback by real-time, haptic) | 6 weeks | Immediate, 1 month | NR |
| **Simic et al, 2012** [12] | Randomised cross-over | 22 (22) | I: 0, II: 9, III: 9, IV: 4, 68.4 (10.2), 27.9 (4.8) | Trunk lean** | A. Trunk lean**: +6 degrees (Real-time, visual) | 1 session | Immediate | None |
| | | | | | B. Trunk lean**: +9 degrees (Real-time, visual) | 1 session | Immediate | None |
| | | | | | C. Trunk lean**: +12 degrees (Real-time, visual) | 1 session | Immediate | None |
| **Simic et al, 2013** [34] | Randomised cross-over | 22 (22) | I: 0, II: 11, III: 6, IV: 5, 69.7 (9), 28.4 (4.8) | Toe-in | A. Toe-in: +10 degrees (Real-time, visual) | 1 session | Immediate | None |
| | | | | | B. FPA: 0 degrees (Real-time, visual) | 1 session | Immediate | None |
| | | | | Toe-out | C. Toe-out: +10 degrees (Real-time, visual) | 1 session | Immediate | None |
| | | | | | D. Toe-out: +20 degrees (Real-time, visual) | 1 session | Immediate | None |
| | | | | | E. Toe-out: +30 degrees (Real-time, visual) | 1 session | Immediate | None |
| **Tokuda et al, 2018** [41] | Pre-post-test | 20 (20) | I: 10, II: 3, III: 4, IV: 3, 72.1 (4.6), 24.0 (2.4) | Trunk lean** | Trunk lean**: +10 degrees (Real-time, visual + clinician verbal) | 1 session | Immediate | NR |

Immediate = immediately following the completion of the program

\* Studies used the feedback as the main strategy aimed to reduce KAM1 to a targeted extent

\*\* Trunk lean indicates ipsilateral trunk lean

\# Participant's gait was retrained after identifying the best method to retrain by computer modelling of the participant's biomechanical data.

† Baseline assessment was done one week before initiating the exercise program.

‡ Outcomes of the program were assessed at week 1 in addition to week 6 (immediately after the program) and the follow-up periods.

KAM1: early stance phase peak external knee adduction moment, FPA: foot progression angle, NR: not reported

**Table 4. Effects of gait modification strategies on indicators of medial knee joint load.**

| Gait modification strategy | Study (Author, year) | Gait implementation | Duration of the program | Assessment timepoint/s | Indicator of medial knee joint load | Baseline value Mean (SD) | Modified gait Mean (SD) |
|---|---|---|---|---|---|---|---|
| **1. Trunk lean** | Gerbrands et al, 2017 [33] | A. Trunk lean: to the greatest possible extent | 1 session | Immediate | KAM1 (Nm/Bw*Ht) | 0.24 (0.12) | 0.15 (0.1) * |
| | | | | | KAM impulse (Nm*s/ Bw*Ht) | 0.08 (0.01) | 0.06 (0.1) * |
| | | | | | KAM2 (Nm/Bw*Ht) | 0.19 (0.12) | 0.15 (0.1) * |
| | | | | | KFM1 (Nm/Bw*Ht) | 0.33 (0.17) | 0.24 (0.2) |
| | | | | | KFM2 (Nm/Bw*Ht) | 0.39 (0.03) | 0.31 (0.04) |
| | Hunt et al, 2011[‡] [31] | A. Trunk Lean: self-selected | 1 session | Immediate | KAM1 (Nm/kg) | 0.81 (0) | 0.38 (0) |
| | Simic et al, 2012[#] [12] | A. Trunk lean: +6 degrees | 1 session | Immediate | KAM1 (Nm/(Bw*Ht) %) | 3.75 (1.06) | 3.4 (1.06) * |
| | | | | | KAM impulse (Nm*s/ (Bw*Ht) %) | 1.22 (0.5) | 1.05 (0.5) * |
| | | | | | KAM2 (Nm/(Bw*Ht) %) | 2.05 (0.83) | 1.71 (0.83) * |
| | | B. Trunk lean: +9 degrees | 1 session | Immediate | KAM1 (Nm/(Bw*Ht) %) | 3.75 (1.06) | 3.33 (1.06) * |
| | | | | | KAM impulse (Nm*s/ (Bw*Ht) %) | 1.22 (0.5) | 1.03 (0.5) * |
| | | | | | KAM2 (Nm/(Bw*Ht) %) | 2.05 (0.83) | 1.69 (0.83) * |
| | | C. Trunk lean: +12 degrees | 1 session | Immediate | KAM1 (Nm/(Bw*Ht) %) | 3.75 (1.06) | 3.19 (1.06) * |
| | | | | | KAM impulse (Nm*s/ (Bw*Ht) %) | 1.22 (0.5) | 0.96 (0.5) * |
| | | | | | KAM2 (Nm/(Bw*Ht) %) | 2.05 (0.83) | 1.56 (0.83) * |
| | Tokuda et al, 2018 [41] | Trunk lean: +10 degrees | 1 session | Immediate | KAM1 (Nm/Kg) | 0.56 (0.21) | 0.41 (0.15) * |
| | | | | | KA/M impulse (Nm*s/ Kg) | 0.19 (0.06) | 0.16 (0.06) * |
| **2. Toe out** | Charlton et al, 2019 [35] | C.Toe out: +10 degrees | 1 session | Immediate | KAM1 (Nm/kg) | 0.48 (0.14) | 0.48 (0.14) |
| | | | | | KAM2 (Nm/kg) | 0.39 (0.14) | 0.37 (0.13) |
| | | D.Toe out: +20 degrees | 1 session | Immediate | KAM1 (Nm/kg) | 0.48 (0.14) | 0.51 (0.14) |
| | | | | | KAM2 (Nm/kg) | 0.39 (0.14) | 0.32 (0.13) |
| | Guo et al, 2017 [37] | Toe out: +15 degrees | 1 session | Immediate | KAM1 (%Bw*Ht) | 2.81 (0.49) | 2.84 (0.44) |
| | | | | | KAM2 (%Bw*Ht) | 2.27 (0.63) | 1.37 (0.53) * |
| | Hunt et al, 2011[‡] [31] | B. Toe-out: self-selected | 1 session | Immediate | KAM1 (Nm/kg) | 0.81 (0) | 0.76 (0) |
| | Simic et al, 2013[#] [34] | C. Toe-out: +10 degrees | 1 session | Immediate | KAM1 (Nm/(Bw*Ht) %) | 3.74 (1.12) | 3.74 (1.12) |
| | | | | | KAM impulse (Nm*s/ (Bw*Ht) %) | 1.23 (0.46) | 1.25 (0.45) * |
| | | | | | KAM2 (Nm/(Bw*Ht) %) | 2.11 (0.77) | 2.09 (0.77) * |
| | | | | | KFM1 (Nm/(Bw*Ht) %) | 2.75 (1.43) | 2.78 (1.43) * |
| | | D. Toe-out: +20 degrees | 1 session | Immediate | KAM1 (Nm/(Bw*Ht) %) | 3.74 (1.12) | 3.92 (1.12) |
| | | | | | KAM impulse (Nm*s/ (Bw*Ht)%) | 1.23 (0.46) | 1.21 (0.45) * |
| | | | | | KAM2 (Nm/(Bw*Ht%) | 2.11 (0.77) | 1.78 (0.77) * |
| | | | | | KFM1 (Nm/(Bw*Ht) %) | 2.75 (1.43) | 2.68 (1.43) * |
| | | E. Toe-out: +30 degrees | 1 session | Immediate | KAM1 (Nm/(Bw*Ht) %) | 3.74 (1.12) | 4.09 (1.12) |
| | | | | | KAM impulse (Nm*s/ (Bw*Ht) %) | 1.23 (0.46) | 1.17 (0.46) * |
| | | | | | KAM2 (Nm/(Bw*Ht) %) | 2.11 (0.77) | 1.36 (0.77) * |
| | | | | | KFM1 (Nm/(Bw*Ht) %) | 2.75 (1.43) | 2.42 (1.48) * |
| | Hunt and Takacs, 2014 [38] | Toe-out: +10 degrees | 10 weeks | Immediate | KAM1 (%Bw*Ht) | 3.45 (0.82) | 3.19 (0.72) |
| | | | | | KAM impulse (% Bw*Ht*s) | 1.33 (0.29) | 1.24 (0.34) |
| | | | | | KAM2 (%Bw*Ht) | 2.87 (0.92) | 2.57 (0.84) * |
| | | | | | KFM1 (%Bw*Ht) | 1.38 (1.36) | 1.51 (1.29) |
| | Hunt et al, 2018[†] [11] | Toe-out: +15 degrees | 4 months | Immediate | KAM1 (%Bw*Ht) | 2.41 (1.33) | 2.43 (0.36) |
| | | | | | KAM impulse (% Bw*Ht*s) | 0.84 (0.44) | 0.82 (0.12) |
| | | | | | KAM2 (%Bw*Ht) | 2.67 (1.2) | 2.44 (0.30) |
| | | | | | KFM1 (%Bw*Ht) | 3.01 (1.45) | 3.14 (0.97) |
| | | | | 1 month | KAM1 (%Bw*Ht) | 2.41 (1.33) | 2.41 (0.41) |
| | | | | | KAM impulse (% Bw*Ht*s) | 0.84 (0.44) | 0.81 (0.12) * |
| | | | | | KAM2 (%Bw*Ht) | 2.67 (1.2) | 2.5 (0.41) * |
| | | | | | KFM1 (%Bw*Ht) | 3.01 (1.45) | 3.38 (0.95) |

*(Continued)*

**Table 4.** (Continued)

| Gait modification strategy | Study (Author, year) | Gait implementation | Duration of the program | Assessment timepoint/s | Indicator of medial knee joint load | Baseline value Mean (SD) | Modified gait Mean (SD) |
|---|---|---|---|---|---|---|---|
| **3. Toe-in** | Booij et al, 2020 [36] | A. Toe-in: individualised to reduce KAM1 by ≥ 10% | 1 session | Immediate | KAM1 (%Bw*Ht) | 2.48 (1.01) | 1.61 (0.93) * |
| | | | | | KFM1 (%Bw*Ht) | 1.70 (3.15) | 1.61 (3.41) |
| | Charlton et al, 2019 [35] | A. Toe-in: +10 degrees | 1 session | Immediate | KAM1 (Nm/kg) | 0.48 (0.14) | 0.40 (0.14) |
| | | | | | KAM2 (Nm/kg) | 0.39 (0.14) | 0.47 (0.13) |
| | | B. FPA: 0 degrees | | | KAM1 (Nm/kg) | 0.48 (0.14) | 0.44 (0.13) |
| | | | | | KAM2 (Nm/kg) | 0.39 (0.14) | 0.42 (0.12) |
| | Shull et al, 2013a [39] | Toe-in: +5 degrees | 1 session | Immediate | KAM1 (%Bw*Ht) | 3.28 (1.37) | 2.9 (1.38) * |
| | | | | | KAM2 (%Bw*Ht) | 1.98 (1.14) | 1.94 (1.09) |
| | | | | | KFM1 (%Bw*Ht) | 1.48 (1.45) | 1.29 (1.39) |
| | | | | | KFM2 (%Bw*Ht) | -1.95 (0.93) | -1.78(1.00) |
| | Simic et al, 2013# [34] | A. Toe-in: +10 degrees | 1 session | Immediate | KAM1 (Nm/(Bw*Ht) %) | 3.74 (1.12) | 3.48 (1.12) * |
| | | | | | KAM impulse (Nm*s/ (Bw* Ht) %) | 1.23 (0.46) | 1.3 (0.46) * |
| | | | | | KAM2 (Nm/(Bw*Ht) %) | 2.11 (0.77) | 2.58 (0.78) * |
| | | | | | KFM1 (Nm/(Bw*Ht) %) | 2.75 (1.43) | 3.32 (1.43) * |
| | | B. FPA: 0 degrees | 1 session | Immediate | KAM1 (Nm/(Bw*Ht) %) | 3.74 (1.12) | 3.65 (1.12) |
| | | | | | KAM impulse (Nm*s/ (Bw* Ht) %) | 1.23 (0.46) | 1.29 (0.45) |
| | | | | | KAM2 (Nm/(Bw*Ht) %) | 2.11 (0.77) | 2.37 (0.78) |
| | | | | | KFM1 (Nm/(Bw*Ht) %) | 2.75 (1.43) | 2.94 (1.43) |
| | Shull et al, 2013b [40] | Toe-in: self-selected | 6 weeks | Immediate | KAM1 (%Bw*Ht) | 3.11 (1.4) | 2.61(1.47) * |
| | | | | | KAM2 (%Bw* Ht) | NR | NS, NR |
| | | | | | KFM1 (%Bw*Ht) | 1.95 (0.76) | 1.67 (0.75) |
| | | | | 1 month | KAM1 (%Bw*Ht) | 3.11 (1.4) | 2.67 (1.41) * |
| | | | | | KAM2 (%Bw* Ht) | NR | NS, NR |
| | | | | | KFM1 (%Bw*Ht) | 1.95 (0.76) | 1.43 (0.70) |
| **4. Medial knee thrust** | Booij et al, 2020 [36] | C. Medial knee thrust: individualised to reduce KAM1 by ≥ 10% | 1 session | Immediate | KAM1 (%Bw*Ht) | 2.48 (1.01) | 1.69 (1.00) * |
| | | | | | KFM1 (%Bw*Ht) | 1.70 (3.15) | 2.39 (3.46) * |
| | Gerbrands et al, 2017 [33] | B. Medial knee thrust: to the greatest possible extent | 1 session | Immediate | KAM1 (Nm/Bw*Ht) | 0.24 (0.12) | 0.17 (0.09) * |
| | | | | | KAM impulse (Nm*s/ Bw*Ht) | 0.08 (0.01) | 0.05 (0.01) * |
| | | | | | KAM2 (Nm/Bw*Ht) | 0.19 (0.12) | 0.17 (0.1) |
| | | | | | KFM1 (Nm/Bw*Ht) | 0.33 (0.17) | 0.15 (0.31) * |
| | | | | | KFM2 (Nm/Bw*Ht) | 0.39 (0.03) | 0.11 (0.04) * |
| | Fregly et al, 2007‡ [30] | Medial knee thrust (trying to walk with old gait pattern) ** | 9 months | Immediate | KAM1 (%Bw*Ht) | 3.8 (0) | 2.3 (0) |
| | | | | | KAM2 (%Bw*Ht) | 4.6 (0) | 2.9 (0) |
| | | Medial knee thrust (trying to walk with modified gait pattern) ** | 9 months | Immediate | KAM1 (%Bw*Ht) | 3.8 (0) | 1.9 (0) |
| | | | | | KAM2 (%Bw*Ht) | 4.6 (0) | 2.1 (0) |
| **5. Medial weight transfer at the foot** | Erhart-Hledik et al, 2017 [16] | A. Medial weight transfer at the foot | 1 session | Immediate | KAM1 (%Bw*Ht) | 2.41 (1.1) | 2.26 (1.04) * |
| | | | | | KAM impulse (% Bw*Ht*s) | 0.77 (0.48) | 0.69 (0.51) * |
| | | | | | KAM2 (%Bw*Ht) | 1.71 (1.01) | 1.47 (0.96) * |
| | | | | | KFM1%Bw*Ht | 2.48 (1.38) | 2.51 (1.42) |
| | | B. Medial weight transfer at the foot: + increased gait speed | 1 session | Immediate | KAM1 (%Bw*Ht) | 2.9 (1.28) | 2.63 (1.35) * |
| | | | | | KAM impulse (% Bw*Ht*s) | 0.71 (0.47) | 0.65 (0.51) * |
| | | | | | KAM2 (%Bw*Ht) | 1.58 (1.11) | 1.5 (1.13) * |
| | | | | | KFM1 (%Bw*Ht) | 3.20 (1.53) | 3.25 (1.79) |
| **6. Wider steps** | Booij et al, 2020 [36] | B. Wider steps: individualised to reduce KAM1 by ≥ 10% | 1 session | Immediate | KAM1 (%Bw*Ht) | 2.48 (1.01) | 1.84 (0.83) |
| | | | | | KFM1 (%Bw*Ht) | 1.70 (3.15) | 1.24 (3.52) |
| **7. Self-selected** | Richards et al, 2018 a [42] | A. Self-selected: to reduce KAM1 by 10% (with real-time visual feedback to reduce KAM1) | 1 session | Immediate | KAM1 (%Bw*Ht) | 3.29 (1) | 3.19 (1.04) |
| | | | | | KAM impulse (% Bw*Ht*s) | 1.11 (0.51) | 1.04 (0.53) |
| | | | | | KFM1 (%Bw*Ht) | 3.15 (1.10) | 3.13 (1.15) |
| | | B. Self-selected: to reduce KAM1 by 10% (with real-time audio feedback to reduce KAM1) | 1 session | Immediate | KAM1 (%Bw*Ht) | 3.29 (1) | 3.18 (0.94) |
| | | | | | KAM impulse (% Bw*Ht*s) | 1.11 (0.51) | 1.08 (0.53) |
| | | | | | KFM1 (%Bw*Ht) | 3.15 (1.10) | 3.16 (1.16) |
| | Cheung et al, 2018 [32] | Self-selected: Adjust either foot progression angle, hip adduction/rotation, and/or trunk sway: to reduce KAM1 by 20% (with real-time visual feedback to reduce KAM1) | 6 weeks | Immediate | KAM1 (Nm/kg*m) | 0.353 (0.053) | 25% significant difference was reported |
| | | | | | KFM1 (Nm/kg*m) | 0.297 (0.0444) | NS, NR |
| | | | | 6 months | KAM1 (Nm/kg*m) | 0.353 (0.053) | 25% significant difference was reported |
| | | | | | KFM1 (Nm/kg*m) | 0.297 (0.0444) | NS, NR |

*(Continued)*

**Table 4.** (Continued)

| Gait modification strategy | Study (Author, year) | Gait implementation | Duration of the program | Assessment timepoint/s | Indicator of medial knee joint load | Baseline value Mean (SD) | Modified gait Mean (SD) |
|---|---|---|---|---|---|---|---|
| 8. Combination of modifications | Richards et al, 2018 a [42] | C. Toe-in, increased step- width and medial knee thrust: to reduce KAM1 by 10% (with real-time visual feedback to reduce KAM1) | 1 session | Immediate | KAM1 (%Bw*Ht) | 3.29 (1) | 2.82 (0.71) * |
| | | | | | KAM impulse (% Bw*Ht*s) | 1.11 (0.51) | 0.89 (0.46) * |
| | | | | | KFM1 (%Bw*Ht) | 3.15 (1.10) | 3.83 (1.49) * |
| | | C. Toe-in, increased step- width and medial knee thrust with visual feedback: to reduce KAM1 by 10% (with real-time visual feedback to reduce KAM1) (Retention (without feedback)) [µ] | 1 session | Immediate | KAM1 (%Bw*Ht) | 3.29 (1) | 3 (0.77) * |
| | | | | | KAM impulse (% Bw*Ht*s) | 1.11 (0.51) | 1.02 (0.47) |
| | | | | | KFM1 (%Bw*Ht) | 3.15 (1.10) | 3.61 (1.48) * |
| | Richards et al, 2018 c [17] | Toe-in +10 degrees (all participants), Step- width: between 15–20 cm (5 out of 21 participants) to reduce KAM1 by 10% (with real-time visual feedback to reduce KAM1) (Natural walking without feedback) [¥] | 6 weeks | Immediate | KAM1 (%Bw*Ht) | 3.65 (0.83) | 3.37 (0.79) * |
| | | | | | KAM impulse (% Bw*Ht*s) | 1.17 (0.33) | 1.15 (0.35) |
| | | | | | KFM1 (%Bw*Ht) | 2.09 (0.85) | 2.14 (0.86) |
| | | | | 3 months | KAM1 (%Bw*Ht) | 3.65 (0.83) | 3.34 (0.76) |
| | | | | | KAM impulse (% Bw*Ht*s) | 1.17 (0.33) | 1.03 (0.33) |
| | | | | | KFM1 (%Bw*Ht) | 2.09 (0.85) | 1.99 (0.78) |
| | | | | 6 months | KAM1 (%Bw*Ht) | 3.65 (0.83) | 3.44 (0.84) |
| | | | | | KAM impulse (% Bw*Ht*s) | 1.17 (0.33) | 1.12 (0.42) |
| | | | | | KFM1 (%Bw*Ht) | 2.09 (0.85) | 2.18 (0.81) |
| | | Toe-in +10 degrees (all participants), Step- width: between 15–20 cm (5 out of 21 participants) to reduce KAM1 by 10% modifying gait: (real-time visual feedback to reduce KAM1) Retention (without feedback) [µ] | 6 weeks | Immediate | KAM1 (%Bw*Ht) | 3.65 (0.83) | 3.31 (0.88) * |
| | | | | | KAM impulse (% Bw*Ht*s) | 1.17 (0.33) | 1.14 (0.37) |
| | | | | | KFM1 (%Bw*Ht) | 2.09 (0.85) | 1.92 (0.85) |
| | | | | Week 1 | KAM1 (%Bw*Ht) | 3.65 (0.83) | 3.14 (0.89) * |
| | | | | | KAM impulse (% Bw*Ht*s) | 1.17 (0.33) | 1.09 (0.38) |
| | | | | | KFM1 (%Bw*Ht) | 2.09 (0.85) | 2.09 (0.91) |

* Significant findings: defined by p ≤ 0.05

[#] Outcomes reported as mean and confidence intervals (CI) (Standard deviations (SD) of these data were calculated using validated statistical methods)

[†] Outcomes reported as mean and Standard Error (SE) (Standard deviations (SD) of these data were calculated using validated statistical methods)

[‡] Case studies (SD of the values are zero)

[**] Outcomes were assessed in 2 different ways: while the participant was trying to walk with the old/baseline gait pattern and trying to walk with a modified gait pattern.

[µ] Assessed the retention effects (without the feedback, though the main strategy is visual feedback)

[¥] The natural walking condition assessed after three- and six-months follow-up period to see the modified gait has been integrated into their everyday gait

KAM1: early stance phase peak external knee adduction moment, KAM2: late stance phase peak external knee adduction moment, KAM: external knee adduction moment impulse, KFM1: early stance phase peak knee flexion moment, KFM2: late stance phase peak knee flexion moment, FPA: foot progression angle

NS: Not significant, NR: Not reported

Bw: Body weight, Ht: Height

Three single-session studies [34, 35, 37] and a 10-week study [38] evaluated immediate effects. A study [11] of a 4-month intervention, evaluated outcomes immediately after the program (at 4 months) and after 1-month follow-up period. From this study, we selected effects immediately after the program. The meta-analyses found toe-out had no effect on KAM1 (ES = 0.02, 95%CI = -0.26, 0.30, k = 5, n = 103 (Fig 3A). Toe-out reduced KAM2 with a small overall effect (ES = -0.42, CI = -0.73, -0.11, k = 5, n = 103), but no dose-response effect was identified (Fig 3B). Toe-out did not change KAM impulse nor KFM1 (ES = -0.12, 95%CI = -0.53, 0.29, k = 3, n = 78 and ES = 0.06, CI = -0.26, 0.38, k = 3, n = 78 respectively) (Fig 3C and 3D). Estimates of $I^2$ ($I^2$ = 0) may be biased as there were very few studies included. No studies reported effects of toe-out on KFM2.

Toe-in reduced KAM1 and increased KAM2 immediately after training with an intervention duration lasting from single-session to 6-weeks. Five studies [34–36, 39, 40] implemented four doses of toe-in (0˚, 5˚,10˚ and individualised) noting that two studies [34, 35] evaluated the same participants in different doses of toe-in. From these studies, we included 10˚ toe-in in the meta-analyses. Four single-session studies [34–36, 39] assessed immediate effects. A study

**A.**

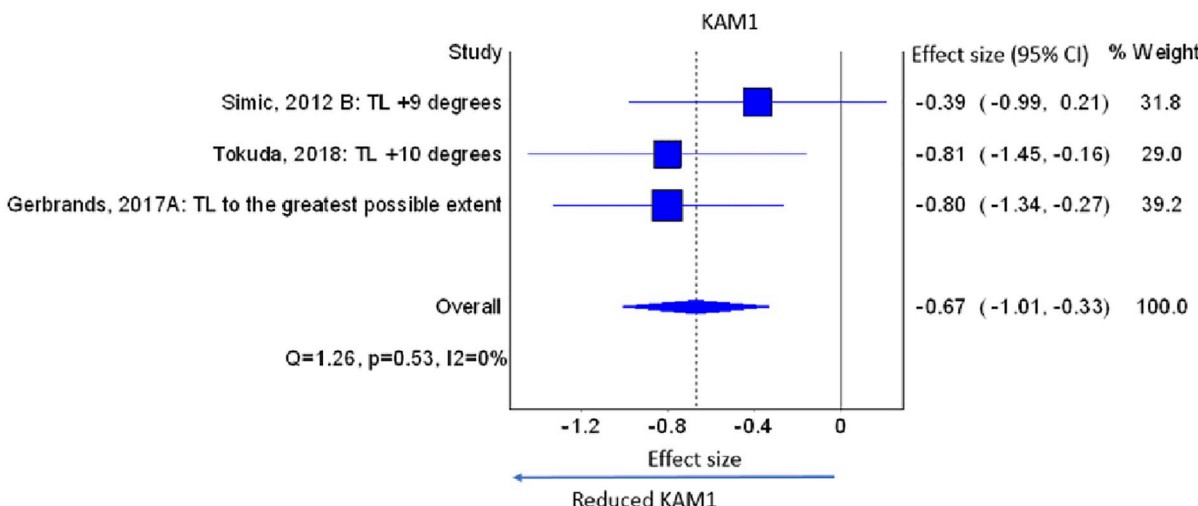

**B.**

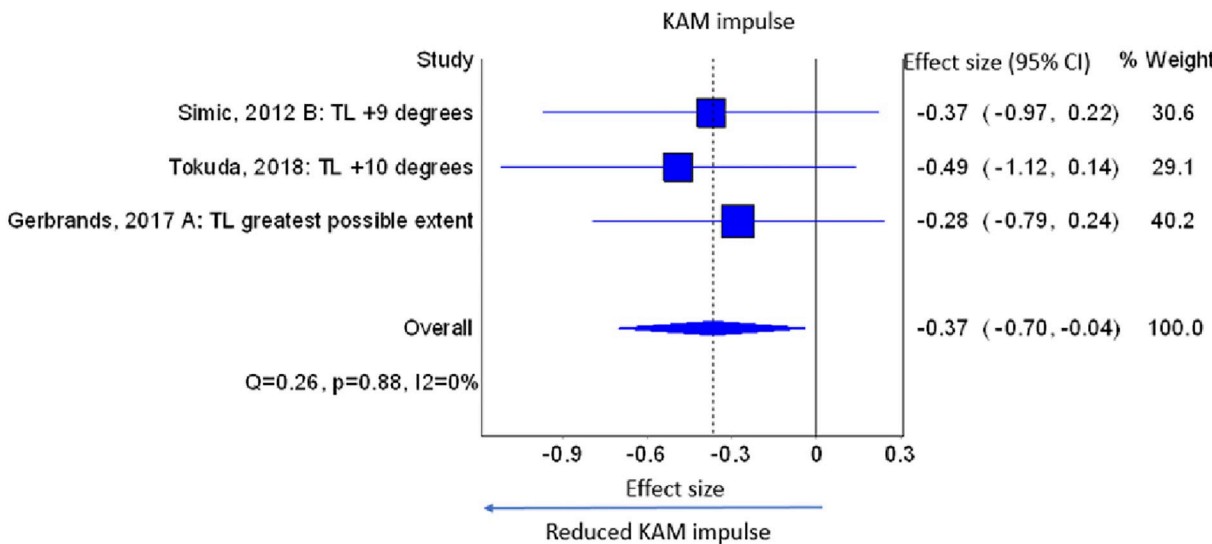

**Fig 2.** A. Effects of trunk lean on KAM1, B. KAM impulse. (TL: Trunk lean, KAM1: early stance phase peak external knee adduction moment, KAM impulse: external knee adduction moment impulse).

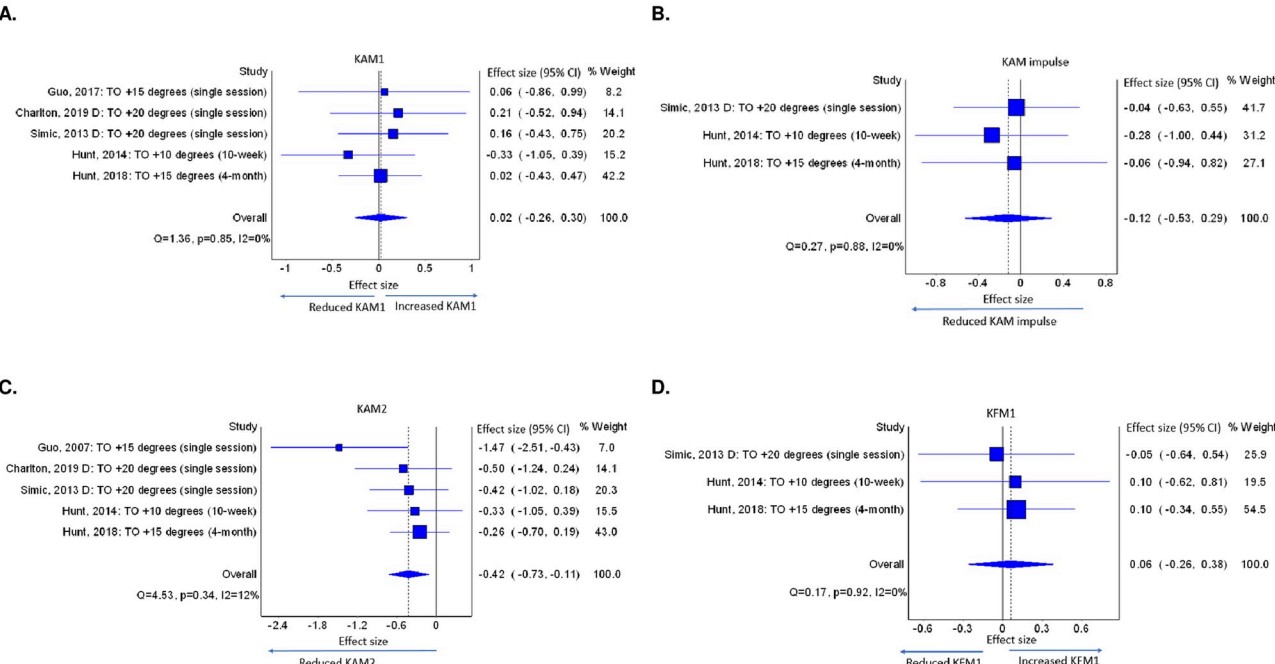

**Fig 3.** A. Effects of toe-out on KAM1, B. KAM impulse, C. KAM2, D. KFM1 (TO: Toe-out; KAM1: early stance phase peak external knee adduction moment, KAM impulse: external knee adduction moment impulse, KAM2: late stance phase peak external knee adduction moment, KFM1: early stance phase peak external knee flexion moment).

[40] with an intervention duration of 6-weeks, assessed outcomes both immediately after the program (at 6 weeks) and again after a one-week follow-up. From this study, we selected effects immediately after the program, for the meta-analyses. Toe-in reduced KAM1 with a medium overall effect (ES = -0.51, 95%CI = -0.81, -0.20, k = 5, n = 89), but no dose-response effect was identified (Fig 4A). Toe-in increased KAM2 with a small overall effect (ES = 0.44, 95% CI = 0.04, 0.85, k = 3, n = 49) and a dose-response effect (larger toe-in produced larger KAM2) (Fig 4B). Toe-in had no effect on KFM1 (ES = 0.04, 95%CI = -.028, 0.37, k = 4, n = 74) (Fig 4C). Estimates of $I^2$ ($I^2$ = 0) may be biased as there were very few studies included. There were insufficient studies to undertake meta-analyses of toe-in on KAM impulse and KFM2.

There were insufficient studies to undertake meta-analyses of interventions targeting medial weight transfer at the foot [16] and wider steps [36]. No meta-analyses of medial knee thrust studies were possible because one of the three studies was a case study [30]. Reported findings suggest these strategies reduce KAM indicators (Table 4). We could not pool the data for studies with specific KAM biofeedback (k = 5) [17, 32, 36, 40, 42], because of the diversity of gait strategies (toe-in, wider steps, self-selected).

### 3.4. Kinematics

Knee joint kinematics were reported in three studies describing four gait strategies (Table 5). Individual studies reported that trunk lean reduced varus angle [31] and increased knee flexion [33]. Medial knee thrust increased knee flexion [33]. Toe-out [31] and medial weight transfer at the foot [16] reduced varus angle.

### 3.5. Adverse effects

While seven studies collected adverse event data (Table 3), three reported none [12, 32, 34] (toe-out, trunk lean and combination of strategies). Four studies reported muscle soreness and

**A.**

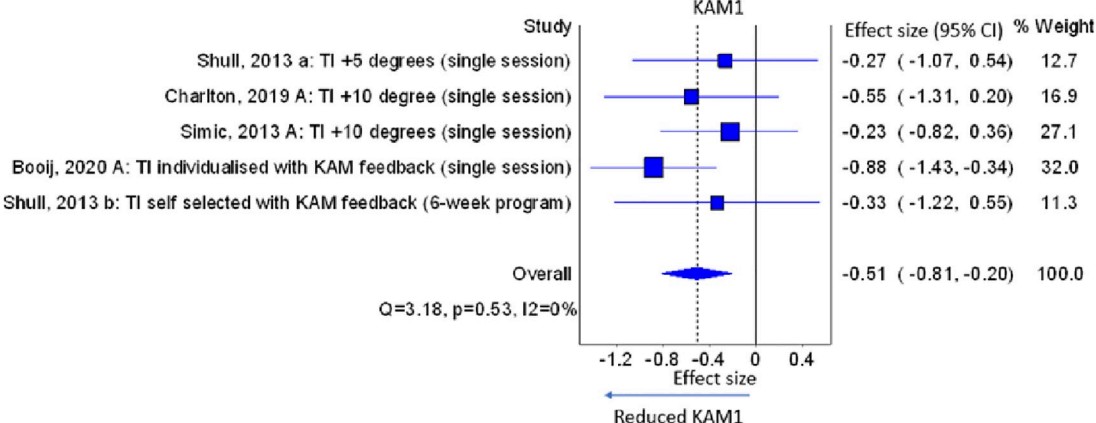

**B.**

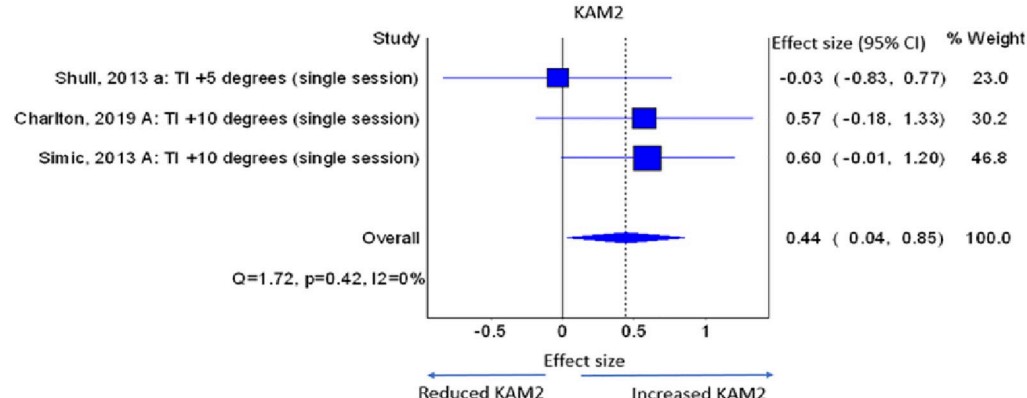

**C.**

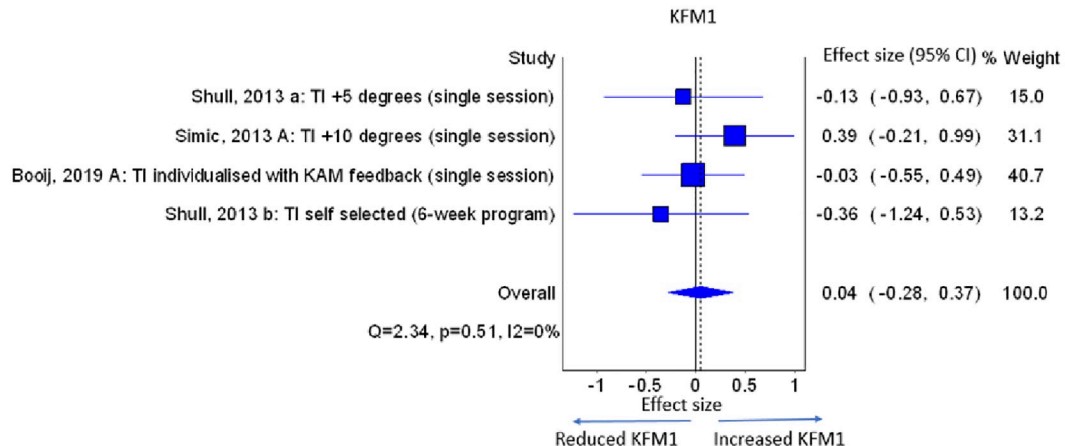

**Fig 4.** A. Effects of toe-in on KAM1, B. KAM2, C. KFM1 (TI: Toe-in, KAM1: early stance phase peak external knee adduction moment, KAM2: late stance phase peak external knee adduction moment, KFM1: early stance phase peak external knee flexion moment).

**Table 5. Effects of gait modifications on the kinematics of the knee joint.**

| Gait modification strategy | Study | Gait modification and implementation | Outcome | Baseline Mean (SD) | Modified gait Mean (SD) |
|---|---|---|---|---|---|
| Trunk lean | Hunt et al, 2011[#] [31] | A. Trunk Lean: self-selected angle | Varus angle (degrees) | 14.3 (0) | 8.9 (0) |
| | Gerbrands, 2017 [33] | A. Trunk lean: to the greatest possible extent | Knee flexion (degrees) | 16.3 (1.8) | 27.0 (3.1) * |
| Toe-out | Hunt et al, 2011[#] [31] | B. Toe-out: self-selected angle | Varus angle (degrees) | 14.3 (0) | 12.5 (0) |
| Medial knee thrust | Gerbrands et al, 2017 [33] | B. Medial knee thrust: to the greatest possible extent | Knee flexion (degrees) | 16.3 (1.8) | 24.2 (2.6) * |
| Medial weight transfer at the foot | Erhart-Hledik et al, 2017 [16] | A. Medial weight transfer at the foot | Varus angle (degrees) | 0.99 (4.9) | 0.29 (4.65) * |
| | | B. Medial weight transfer at the foot + increased gait speed | Varus angle (degrees) | 1.33 (4.79) | 0.75 (4.98) * |

* Significant findings defined by p ≤ 0.05

[#] A case study, therefore, SD of the values are zero

joint pain, but, symptoms resolved during the training time [11, 17, 31, 38] (toe-out and a specific KAM biofeedback study).

## 3.6. Certainty of evidence

Certainty of evidence according to the GRADE approach [28, 29] for the efficacy of ipsilateral trunk lean, toe-out and toe-in on medial knee joint load reduction was very-low to low (Table 6). The overall risk of bias was downgraded by one level because the risk of bias of the studies was good to fair. No serious inconsistency was found. No serious indirectness was identified, and the evidence was appropriate for the population. The potential for imprecision was downgraded by one level because of the limited number of studies and small sample sizes. Publication bias was not found. A dose-response effect for the extent of trunk lean to reduce KAM1 upgraded the certainty of evidence. No dose-response effect was found for trunk lean on KAM impulse, nor toe-out on KAM2. Among toe-in studies, no dose-response effect was identified on KAM1, but a dose-response effect was identified where toe-in increased KAM2.

## 4. Discussion

This systematic review evaluated the effects of neuromuscular gait modification strategies on indicators of medial knee joint load exclusively in participants with medial knee OA. We found with very-low to low certainty of evidence that ipsilateral trunk lean reduced KAM1 (medium effect) and KAM impulse (small effect) and toe-out reduced KAM2 (small effect). Toe-in reduced KAM1 (medium effect) but increased KAM2 (small effect). Our findings support three previous systematic reviews [13–15]. Simic et al. in 2011 suggested that trunk lean consistently reduced KAM1 and toe-out consistently reduced KAM2 [13]. Bowd et al. in 2019 reported that trunk lean resulted in a larger reduction in KAM1 [14]. Wang et al. in 2020 concluded that toe-out reduced KAM2 [15]. Our study builds on these previous reviews including meta-analyses to determine the efficacy of each gait modification exclusively in people with medial knee OA.

Ipsilateral trunk lean reduced KAM1 and reduced KAM impulse with a dose-response effect. Although the optimum dose of trunk lean remains unknown, a dose-response would suggest that maximum trunk lean is likely to be most effective. Participants were able to

**Table 6. The quality of evidence on indicators of medial knee joint load assessed according to the GRADE approach.**

| Outcomes | Number of studies (participants) | Study design (number of studies) | Risk of bias* | Inconsistency | Indirectness | Imprecision | Publication bias | Overall effects (ES (95% CI) | Dose-response effect | Certainty of the evidence (GRADE) |
|---|---|---|---|---|---|---|---|---|---|---|
| **Effects of trunk lean** | | | | | | | | | | |
| **Effects of trunk lean on KAM1** | 3 (72) | Randomised cross-over (2), pre-post-test (1) | Serious | Not serious | Not serious | Serious† | Undetected# | -0.67 (-1.01 to 0.33) | Yes | ⊕⊕◯◯ LOW |
| **Effects of trunk lean on KAM impulse** | 3 (72) | Randomised cross-over (2), pre-post-test (1) | Serious | Not serious | Not serious | Serious† | Undetected# | -0.37 (-0.7 to 0.04) | No | ⊕◯◯◯ VERY-LOW |
| **Effects of toe-out** | | | | | | | | | | |
| **Effects of toe-out on KAM2** | 5 (103) | RCT (1), Randomised cross-over (2), pre-post-test (2) | Serious | Not serious | Not serious | Serious† | Undetected# | -0.42 (-0.73 to 0.11) | No | ⊕◯◯◯ VERY -LOW |
| **Effects of toe-in** | | | | | | | | | | |
| **Effects of toe-in on KAM1** | 5 (89) | Randomised cross-over (2), pre-post-test (3) | Serious | Not serious | Not serious | Serious† | Undetected# | -0.51 (-0.81 to 0.20) | No | ⊕◯◯◯ VERY-LOW |
| **Effects of toe-in on KAM2** | 3 (49) | Randomised cross-over (2), pre-post-test (1) | Serious | Not serious | Not serious | Serious† | Undetected# | 0.44 (0.04 to 0.85) | Yes | ⊕⊕◯◯ LOW |

KAM1 (early stance phase peak external knee adduction moment), KAM impulse (external knee adduction moment impulse), KAM2 (late stance phase peak external knee adduction moment)

* Risk of bias of studies reported in an additional table (Table 2)

# There should be at least 7 studies to evaluate the publication bias. Though lack of studies, publication bias was undetected and not downgraded the quality

† Downgraded for imprecision due to limited number of studies and small sample size

LOW certainty of evidence: This research provides some indication of the likely effect. However, the likelihood that it will be substantially different (a large enough difference that it might have an effect on a decision) is high

VERY-LOW certainty of evidence: This research does not provide a reliable indication of the likely effect. The likelihood that the effect will be substantially different (a large enough difference that it might have an effect on a decision) is very high

achieve a trunk lean of 12° with real-time visual feedback [12]. No adverse events were reported from trunk lean, yet potentially shifting load from the medial to the lateral tibiofemoral compartment should be considered. These findings were based on single-session studies, therefore, clinical recommendations are made with caution since the longer-term effects (beyond single session) of trunk lean are unknown.

Toe-out reduced KAM2 but had no effect on KAM1. Individual studies showed a dose-response effect where larger toe-out produced greater KAM2 reductions [34, 35]. Participants achieved up to 20° toe-out with visual feedback [34]. Though longer-term studies (10-weeks and 4-months) reported joint pain, this resolved with training. Although only reported in an observational study, participants who naturally walked with a toe-out gait pattern [43], had less medial knee OA structural disease progression over 18 months compared to those who walked with a more neutral FPA, suggesting that larger toe-out is associated with lower knee OA disease progression. Future studies are needed to evaluate this in a controlled trial environment.

Toe-in reduced KAM1 and while it appeared that a minimum of 5˚ toe-in was critical, the most important factor was feedback. Larger toe-in produced larger KAM1 reductions [34, 35] with the highest KAM1 reduction achieved by individualised FPA, adjusted using specific KAM biofeedback [36]. The systematic review by Wang et al (2020) found a KAM1 reduction in healthy people, but not those with knee OA. But since that review, the Booij study has shown that 5˚ toe-in can be effective when specific KAM biofeedback is provided [39, 44]. When applying these gait interventions in clinical practice, it should be considered that toe-in reduces KAM1 but increases KAM2. However, reducing KAM1 is more important because KAM1 is associated with OA progression [45].

We were not able to conduct meta-analyses on several strategies due to the limited available studies. There is potential that real-time feedback emerges as a powerful gait modification strategy, particularly as it permits patients to self-select effective strategies. Five specific KAM biofeedback studies included in this review demonstrated a reduction of knee joint load indicators. However, these studies used a diverse range of strategies, including toe-in, wider steps and self-selected strategies, so they could not be pooled. Since our search, an RCT [46] of a 6-week sensor-based gait training program has also shown an effective reduction in KAM1, when participants were asked to adjust their FPA using specific KAM biofeedback. There has been an additional single-session cohort study using medial knee thrust [47] which demonstrates an ability to reduce KAM1, further supporting the potential of medial knee thrust gait. Gait retraining using real-time feedback seems to be the way forward in the reduction of knee joint load effectively.

Adverse events have not been an issue with the gait interventions. Some joint pain or muscle fatigue resolved with training. This agrees with a systematic review that found no evidence of increased joint load on hip, ankle or spine from gait modifications to unload the knee [14], but there are still very few studies that capture longer-term effects.

The results of this systematic review should be considered in light of its limitations. Studies with gait aids and orthoses, which are extrinsic interventions were beyond the scope of this review. We did not evaluate the effects of gait modification on pain and function. Further, we focused on studies with medial knee OA, and hence, findings will not transfer to people with OA in other compartments of the knee. Even though we adjusted the meta-analyses according to study quality, there were few randomised studies available. Moreover, the results of the meta-analyses of this review potentially have been affected by clinical heterogeneity of included studies; including treatment duration, dose, and whether specific KAM feedback is used. The main source of clinical heterogeneity is probably intervention time. However, the majority of studies reported single-session data, and the forest plots did not suggest that intervention time significantly altered the effect size. Therefore, the clinical implication is that a single session is likely to be effective, however, the longer-term effects of the strategies are not yet well understood. The effects of trunk lean and toe-out seem to be enhanced by larger angles and the effect of toe-in is likely enhanced by specific KAM feedback. Therefore, when applying the findings of this review to practice, clinical heterogeneity in intervention suggests that trunk lean and toe-out with greater angles, and toe-in with specific KAM feedback may be more effective.

Although more work is needed to form clinical recommendations, gait strategies including ipsilateral trunk lean, toe-out and toe-in have a potential to reduce indicators of knee joint load in people with medial knee OA. Medial knee thrust, medial weight transfer and wider steps may also reduce knee load based on results of individual studies. Greater trunk lean, larger toe-in or toe-out produce greater knee joint load reductions. While the certainty of the evidence is very-low to low, there is potential for these interventions to be clinically helpful. Included studies demonstrated that participants were capable of achieving peak trunk lean of

12˚, toe-out of 20˚, and toe-in of 10˚. However, future research is required to determine the optimum angles to develop clinical recommendations. Feedback (visual, verbal or haptic) is necessary to train gait strategies. Adverse events have not been an issue, but there are still few studies that capture longer-term effects. Therefore, future longer-term studies are recommended assessing knee joint load along with pain, function and adverse effects.

## Supporting information

**S1 Appendix. Search strategy (in MEDLINE database).**
(DOCX)

## Author Contributions

**Conceptualization:** M. Denika C. Silva, Diana M. Perriman, Angela M. Fearon, Milena Simic, Rana S. Hinman, Kim L. Bennell, Jennie M. Scarvell.

**Data curation:** M. Denika C. Silva, Diana M. Perriman.

**Formal analysis:** M. Denika C. Silva.

**Investigation:** M. Denika C. Silva, Daniel Tait, Trevor J. Spencer.

**Methodology:** M. Denika C. Silva, Diana M. Perriman, Angela M. Fearon, Dianne Walton-Sonda, Milena Simic, Rana S. Hinman, Kim L. Bennell, Jennie M. Scarvell.

**Project administration:** M. Denika C. Silva, Jennie M. Scarvell.

**Supervision:** Diana M. Perriman, Angela M. Fearon, Jennie M. Scarvell.

**Validation:** M. Denika C. Silva, Diana M. Perriman.

**Visualization:** M. Denika C. Silva, Diana M. Perriman.

**Writing – original draft:** M. Denika C. Silva.

**Writing – review & editing:** M. Denika C. Silva, Diana M. Perriman, Angela M. Fearon, Daniel Tait, Trevor J. Spencer, Dianne Walton-Sonda, Milena Simic, Rana S. Hinman, Kim L. Bennell, Jennie M. Scarvell.

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
