## [Decision Letter · Decision Letter 0]

5 Apr 2022

PONE-D-21-38506Effects of neuromuscular gait modification strategies  on indicators of knee joint load in people with medial knee osteoarthritis: A systematic review and meta-analysisPLOS ONE

Dear Dr. Silva,

Thank you for submitting your manuscript to PLOS ONE. After careful consideration, we feel that it has merit but does not fully meet PLOS ONE’s publication criteria as it currently stands. Therefore, we invite you to submit a revised version of the manuscript that addresses the points raised during the review process.

We look forward to receiving your revised manuscript.

Kind regards,

Daniel Ribeiro

Academic Editor

PLOS ONE

Journal Requirements:

No

Rana S Hinman is supported by an NHMRC Senior Research Fellowship (#1154217). Kim L Bennell is supported by an NHMRC Investigator Grant (#1174431). Milena Simic is supported by the Sydney University SOAR fellowship. Kim L Bennell receives personal fees from Wolters Kluwer for UpToDate knee OA clinical guidelines. There are no further conflicts of interest to disclose.

Additional Editor Comments:

Thank you for your submitting your work to PLOS One.

First, my apologies for the long time to process your submission. I tried my best to secure 2 reviewers to assess your submission, but I was not successful with it. A large number of colleagues rejected the invite due to workload - I imagine the disruptions of COVID-19 on workload, Christmas break and the summer break (for colleagues in the southern hemisphere) were the main causes for the challenges I faced when recruiting reviewers.

I managed to secure one reviewer who is an expert in the field and provided feedback on your manuscript. Below, my feedback in addition to those submitted by the reviewer. Most of my feedback and suggestions aim to improve clarity and coherence of your manuscript.

Line 33, “Findings have very low to low certainty of evidence”: suggest re-wording. The level of certainty is an interpretation that we, researchers, make. It does not “belong” to the findings we observe.

Line 34 to 37, Conclusions: I suggest you revising it to consider the certainty of evidence when presenting your conclusions. In the current version, it seems you have a definite answer supporting strategies such as ipsilateral trunk and toe-off.

Introduction

Lines 45 to 47, “As knee OA…”: this reads like the aim of the review. I suggest revising the intro for flow, so that the rationale is presented for the reader first and at the end of the introduction, you describe the focus of your review.

Line 64, “;”: suggest you replacing semi-colon with colon, since you are listing gait modification strategies.

Lines 61 to 71, “These strategies…”: this sentence could be revised to improve flow and clarity. What do you mean by “systematically across all the indicators of load”?

Line 77, “specific feedback”: what do you mean by specific feedback?

Lines 81 to 83, “Our systematic review builds on these … OA”: this paragraph could be revised to improve clarity and flow. Your concluding sentence refers to “all gait modification strategies” but the supporting sentences refer to review by Simic that included people without knee OA, feedback strategies (i.e. haptic sensors and specific feedback), a review that excluded papers assessing knee moments, and another review that focused on toe-out and toe-in strategies. I suggest you revising it to improve the coherence of the message being presented for the reader and making it explicit for the reader what your review is adding that those previous reviews did not cover.

Lines 85 to 87: your aim is slightly different to the research question described on PROSPERO. In your manuscript, you seem to have adopted a narrower aim. I suggest you to revise this to ensure your manuscript reflects what you described in the protocol.

Literature search: can I please suggest you update your search, given it is now more than 12 months old?

Line 107: I suggest you present in more detail the inclusion and exclusion criteria used for screening studies. I noticed you refer to S2 Appendix, but I did not have access to that info. Given the relevance of inclusion and exclusion criteria, I suggest you describe that in the main manuscript or present it as a table.

Later in your methods (line 154) you present a definition for gait strategies. I suggest you describing this earlier, after you list the inclusion and exclusion criteria or in section 2.4 (as the first info you present in that section). That will help the reader to understand what you mean by the term gait strategies in your review.

Line 109: in your protocol you were more specific about the type of design you considered for your review.

Line 117, “conflicts were resolved by a third reviewer”: minor comment. Should you edit this to “resolved by consulting with a third reviewer”?

Line 133, “;”: replace with colon, giving you are listing data extracted.

Line 136 to 137, “kinematic data”: should you be more specific here? Kinematic data involves position, velocity, acceleration, angles, etc. Did you extract any data, all kinematic data reported or joint angles only?

Lines 141 to 142, “the effects of gait modification strategies”: should you revise this sentence? The effect of gait mod strategies on what?

Line 144, “studies of a similar gait modification strategy”: should you revise this for clarity? Studies ‘using/testing’(?) similar gait modification strategies?

Lines 144 to 145: apologies, but what do you mean by “single case study”? If you were not planning to include in the analysis, should you have used this design as exclusion criteria?

Lines 157 to 158 “dose that was most commonly used”: most commonly used by whom? By included studies?

Line 159 “we selected one dose”: could revise for clarity. In this explanation, you do not refer to “most commonly used” and it suggests you one at your own will.

Line 163 “power”: is power the most appropriate term or is it size?

Lines 164 and 165: so how did you assess heterogeneity in your analyses? You seem to be addressing here only the statistical heterogeneity (as per this and your results section). You could have discussed the likely impact of heterogeneity between included studies from a clinical and methodological point of view.

Line 176, “was used”: past tense. Previous verb in the sentence using present tense.

Line 177, “we used the lower to indicate the risk of bias”: revise for clarity.

Line 191, “good to fair”: you did not present the criteria to categorize Qi scores, so your statement that quality of studies ranged from good to fair is mismatched with the findings presented in brackets. That makes interpretation of your findings implicit.

Lines 216 to 217: minor comment. Is “communicate” the best term here? These feedback methods are used as interventions.

Table 2: apologies but it is not clear what you mean by “I: 12, II: 7, III: 7, IV: 4”

Lines 273 to 274, “Heterogeneity values (I2) were estimated but not reliable (k=3)”: apologies, but I struggled to follow this. Heterogeneity is not a measure of reliability. Can I suggest you to re-word this to improve clarity and flow? There are other sections in your manuscript in which you reported heterogeneity in a similar way.

Line 378, “We found with very low”: with – typo?

Line 406, “may have an impact”: I suggest you being more specific. This is a generic statement. Having an impact could be for the better or worse.

Line 415, “In clinical practice, consider that toe-in”: please revise sentence for flow and clarity.

Line 434, “can reduce indicators of knee”: is it appropriate to state that? Your findings suggest very low to low certainty of evidence, with most interventions having a small effect. Taking these into account, I wonder whether your statement is too optimistic and supportive of those interventions. Later in that same paragraph, you adopt a more cautious interpretation (e.g. “While the certainty of the evidence is very low to low, there is potential for these interventions to be clinically helpful.”). my suggestion it to keep this cautious way of interpreting data throughout the manuscript.

Reviewers' comments:

Reviewer's Responses to Questions

**Comments to the Author**

1. Is the manuscript technically sound, and do the data support the conclusions?

Reviewer #1: Yes

2. Has the statistical analysis been performed appropriately and rigorously? 

Reviewer #1: Yes

3. Have the authors made all data underlying the findings in their manuscript fully available?

Reviewer #1: Yes

4. Is the manuscript presented in an intelligible fashion and written in standard English?

Reviewer #1: Yes

5. Review Comments to the Author

Reviewer #1: This was a systematic review and meta-analysis examining the effect of gait modifications on biomechanical indicators of medial knee joint load. This paper was very well written and clear. The paper met all of the criteria for publication in PLOS ONE; the paper presented the results of primary scientific research, the results have not been previously published, the methodology are appropriate and described in excellent detail, conclusions are supported by the data, and data used in the meta-analysis are available in the tables. This paper will be relevant for clinicians as it details easy-to-implement gait modifications for people with medial compartment knee osteoarthritis, and the evidence for their effectiveness. I only have two minor comments/suggestions:

In some of the Tables there are some instances of NM rather than Nm. Please correct.

Line 308-309: “A study which intervention lasting for 6-weeks…”- reword

6. PLOS authors have the option to publish the peer review history of their article (what does this mean?). If published, this will include your full peer review and any attached files.

Reviewer #1: No

---

## [Author Response · Author response to Decision Letter 0]

23 May 2022

Rebuttal in response to reviewers

Revision- PONE-D-21-38506- Effects of neuromuscular gait modification strategies on indicators of knee joint load in people with medial knee osteoarthritis: A systematic review and meta-analysis.

Author’s name: M Denika C Silva email: denika.silva@canberra.edu.au

We greatly appreciate your time in reviewing the manuscript and the detailed comments provided which are helpful to improve the paper.

Please see the following point-by-point responses to each comment raised by the academic editor and reviewer (s).

Editor and reviewers’ comments are in bold italic followed by the author's responses in blue font.

Please note the line numbers and page that we mention in this reply, refer to the manuscript with track changes.

Journal Requirements

1. Please ensure that your manuscript meets PLOS ONE's style requirements

This has been completed. Thank you.

2. Thank you for stating the following financial disclosure: At this time, please address the following queries:

The authors received no specific funding for this work. 

The authors received no specific funding for this work. 

The authors received no specific funding for this work.

The authors received no specific funding for this work. The statement has been added to the cover letter. Thank you.

Rana S Hinman is supported by an NHMRC Senior Research Fellowship (#1154217). Kim L Bennell is supported by an NHMRC Investigator Grant (#1174431). Milena Simic is supported by the Sydney University SOAR fellowship. Kim L Bennell receives personal fees from Wolters Kluwer for UpToDate knee OA clinical guidelines. There are no further conflicts of interest to disclose.

All the competing interests are included in the cover letter. We included the statement “This does not alter our adherence to all PLOS ONE policies on sharing data and materials”. in the cover letter. Please update the online submission. 

All the data are included in the manuscript and supplements, and no additional data are provided in a repository. Please update the data availability statement. The statement has been added to the cover letter.

This study is a systematic review and does not involve human participants. Therefore, ethics approval is not applicable.

We reviewed the reference list and can confirm that it is complete and correct. The reference list does not include any papers that have been retracted.

 

Additional Editor Comments

Line 33, “Findings have very low to low certainty of evidence”: suggest re-wording. The level of certainty is an interpretation that we, researchers, make. It does not “belong” to the findings we observe.

Thank you. This has been revised. This now reads “Certainty of evidence was very-low to low according to the GRADE approach” (lines 33-35, track changes copy)

Line 34 to 37, Conclusions: I suggest you revising it to consider the certainty of evidence when presenting your conclusions. In the current version, it seems you have a definite answer supporting strategies such as ipsilateral trunk and toe-off.

This has been revised. Please see lines 40-42. This now reads “Very-low to low certainty of evidence suggests that there is a potential that ipsilateral trunk lean, toe-out, and toe-in to be clinically helpful to reduce indicators of medial knee joint load. There is yet little evidence for interventions over several weeks.”

Introduction

Lines 45 to 47, “As knee OA…”: this reads like the aim of the review. I suggest revising the intro for flow, so that the rationale is presented for the reader first and at the end of the introduction, you describe the focus of your review.

This paragraph has been revised. Please see lines 45-52. This now reads “Non-surgical management strategies for knee osteoarthritis (OA) have become a high priority with increasing prevalence [1, 2]. Besides, knee OA commonly occurs in the medial compartment of the joint [3]. As new strategies and programs emerge, a comprehensive understanding of which non-surgical strategies have the potential for arresting or slowing knee OA progression is urgently required [2]. Gait retraining may have the potential to slow disease progression by reducing knee joint load since knee joint load is associated with the progression of medial knee OA [4, 5].

Line 64, “;”: suggest you replace semi-colon with colon, since you are listing gait modification strategies.

Thank you. This has been amended and replaced with a colon (line 71).

Lines 61 to 71, “These strategies…”: this sentence could be revised to improve flow and clarity. What do you mean by “systematically across all the indicators of load”?

This has been revised and now reads “These strategies have demonstrated some ability to reduce the indicators of medial knee joint load, such as KAM. However, a comprehensive synthesis of gait modifications would assist their implementation in clinical practice”. Please see lines 76-80.

Line 77, “specific feedback”: what do you mean by specific feedback?

This has been revised. This now reads “Recently, the field has advanced to incorporate innovative feedback strategies such as haptic sensors [16] and real-time feedback on knee joint moments [17].” (line 92-94).

Lines 81 to 83, “Our systematic review builds on these … OA”: this paragraph could be revised to improve clarity and flow. Your concluding sentence refers to “all gait modification strategies” but the supporting sentences refer to review by Simic that included people without knee OA, feedback strategies (i.e. haptic sensors and specific feedback), a review that excluded papers assessing knee moments, and another review that focused on toe-out and toe-in strategies. I suggest you revising it to improve the coherence of the message being presented for the reader and making it explicit for the reader what your review is adding that those previous reviews did not cover.

This paragraph has been revised. This now reads “Three previous systematic reviews have analysed the efficacy of gait modification strategies on medial knee joint load [13-15]. In 2011, Simic et al. reviewed 24 gait retraining studies, fourteen of which investigated healthy participants without knee OA [13]. A review by Bowd et al. (2019) specifically investigated whether gait modifications aimed at negative consequences for loads at the hip and ankle. [14]. Wang et al. (2020) investigated effects of toe-out and toe-in strategies [15] but did not explore any of the other common strategies such as trunk lean, medial knee thrust, etc. Recently, the field has advanced to incorporate innovative feedback strategies such as haptic sensors [16] and real-time feedback on knee joint moments [17]. Our systematic review adds to previous reviews by including all neuromuscular gait modification strategies and exclusively in people with medial knee OA.” (lines 82-96).

Lines 85 to 87: your aim is slightly different to the research question described on PROSPERO. In your manuscript, you seem to have adopted a narrower aim. I suggest you to revise this to ensure your manuscript reflects what you described in the protocol.

This has been revised and included the research question described on PROSPERO. This now reads “the aim of this systematic review was to determine the effects of neuromuscular gait modification strategies on indicators of medial knee joint load in people with medial knee OA.” (lines 98-101).

Literature search: can I please suggest you update your search, given it is now more than 12 months old?

We updated the search and screened 360 records and ended up with two eligible papers. These two studies were added to the discussion section. Please see lines 457-463.

“Since our search, an RCT [46] of a 6-week sensor-based gait training program has also shown an effective reduction in KAM1, when participants were asked to adjust their FPA using specific KAM biofeedback. There has been an additional single-session cohort study using medial knee thrust [47] which demonstrates an ability to reduce KAM1, further supporting the potential of medial knee thrust gait. Gait retraining using real-time feedback seems to be the way forward in the reduction of knee joint load effectively.”

46. Wang S, Chan PPK, Lam BMF, Chan ZYS, Zhang JHW, Wang C, et al. Sensor-Based Gait Retraining Lowers Knee Adduction Moment and Improves Symptoms in Patients with Knee Osteoarthritis: A Randomized Controlled Trial. Sensors. 2021;21(16).

47. Bokaeian HR, Esfandiarpour F, Zahednejad S, Mohammadi HK, Farahmand F. Effects of Medial Thrust Gait on Lower Extremity Kinetics in Patients with Knee Osteoarthritis. Ortop Traumatol Rehabil. 2021;23(2):115-20.

Line 107: I suggest you present in more detail the inclusion and exclusion criteria used for screening studies. I noticed you refer to S2 Appendix, but I did not have access to that info. Given the relevance of inclusion and exclusion criteria, I suggest you describe that in the main manuscript or present it as a table.

Thank you. Inclusion and exclusion criteria are now presented as a table in the manuscript (table 1). The description is revised in line 126 (mentioned as ‘table 1’).

Table 1: Criteria for the eligibility of papers included in the systematic review

Inclusion criteria

1. Any study design (e.g. randomised controlled trials, quasi clinical trials, cohort studies, case series, studies with or without a control group)

2. Adults aged 18 years or older

3. Medial compartment knee osteoarthritis confirmed by imaging

4. Any intervention where the participants are taught a new walking pattern that is aimed at reducing the load on the medial compartment of the knee and its effects can be determined in isolation from other intervention effects.

5. Within-subject measures of gait before and after intervention were recorded

6. Outcomes were indicators of medial knee joint load

Exclusion criteria

1. No original data (e.g. a review or editorial)

2. Abstracts only and other materials not published as a full peer-reviewed paper

3. Predominantly lateral compartment knee osteoarthritis

4. Predominantly patellofemoral knee osteoarthritis

5. Concurrent osteoarthritis in other lower limb joints unless data are reported separately

6 Interventions with gait aids or orthoses

7. Intervention effects cannot be determined in isolation from other intervention effects

Later in your methods (line 154) you present a definition for gait strategies. I suggest you describing this earlier, after you list the inclusion and exclusion criteria or in section 2.4 (as the first info you present in that section). That will help the reader to understand what you mean by the term gait strategies in your review.

This has been revised and the definition for gait strategies is included (moved to) as the first info in section 2.4 (lines 162-165).

Line 109: in your protocol you were more specific about the type of design you considered for your review.

Thank you, this is included in Table 1 (line 133, page 6 and 7)

Line 117, “conflicts were resolved by a third reviewer”: minor comment. Should you edit this to “resolved by consulting with a third reviewer”?

This has been revised. This now reads “conflicts were resolved by consulting with a third reviewer” (line 131).

Line 133, “;”: replace with colon, giving you are listing data extracted.

This has been revised and replaced with a colon (line 150).

Line 136 to 137, “kinematic data”: should you be more specific here? Kinematic data involves position, velocity, acceleration, angles, etc. Did you extract any data, all kinematic data reported or joint angles only?

This has been revised. This now reads “Secondary outcomes were 3D knee kinematic data measuring flexion-extension, abduction-adduction, and internal-external rotation angles” (lines 154-155).

Lines 141 to 142, “the effects of gait modification strategies”: should you revise this sentence? The effect of gait mod strategies on what?

This has been revised. This now reads “effects of gait modification strategies on indicators of medial knee joint load (lines 161-162).

Line 144, “studies of a similar gait modification strategy”: should you revise this for clarity? Studies ‘using/testing’(?) similar gait modification strategies?

This has been revised and now reads “studies using similar gait modification strategies” (lines 166-167).

Lines 144 to 145: apologies, but what do you mean by “single case study”? If you were not planning to include in the analysis, should you have used this design as exclusion criteria?

There were two studies with a single case study design. They were included in the descriptive synthesis in table 2, 3, 4 and 5. But they were not included in the meta-analyses because of the potential for bias and effect size cannot be calculated for meta-analyses. This statement is added now in the methods section. This now reads “Single case studies were not included in the meta-analyses because of the potential for bias and effect size cannot be calculated.” (lines 167-168).

Lines 157 to 158 “dose that was most commonly used”: most commonly used by whom? By included studies?

This has been revised. This now reads “Where a single study reported multiple doses of the same gait strategy, a single representative dose that was most commonly used by included studies was selected for the meta-analysis” (lines 179-181)

Line 159 “we selected one dose”: could revise for clarity. In this explanation, you do not refer to “most commonly used” and it suggests you one at your own will.

This has been revised. This now reads “we selected a single representative dose that was most commonly used by included studies, and the overall effect was calculated from that data” (lines 182-184).

Line 163 “power”: is power the most appropriate term or is it size?

This has been revised and amended with the word “effect size” (line 188).

Lines 164 and 165: so how did you assess heterogeneity in your analyses? You seem to be addressing here only the statistical heterogeneity (as per this and your results section). You could have discussed the likely impact of heterogeneity between included studies from a clinical and methodological point of view.

This has been revised and amended as statistical heterogeneity (I2) values” (lines 189, 299, 322 and 344).

Line 176, “was used”: past tense. Previous verb in the sentence using present tense.

Thank you. This has been revised. This now reads “is used” (line 201)

Line 177, “we used the lower to indicate the risk of bias”: revise for clarity.

This has been revised. This now reads “Where there were several study designs, we considered the certainty of evidence is low due to methodological heterogeneity.” (lines 202-204).

Line 191, “good to fair”: you did not present the criteria to categorize Qi scores, so your statement that quality of studies ranged from good to fair is mismatched with the findings presented in brackets. That makes interpretation of your findings implicit.

This has been revised and discussed using the total points. This now reads “The overall quality of studies was good to fair (total point scored, mode = 19, range 13 to 24) except for two case studies [30, 31] (Table 2).” (lines 217-221).

Lines 216 to 217: minor comment. Is “communicate” the best term here? These feedback methods are used as interventions.

This has been revised. This now reads “A variety of feedback methods were used in included studies, for example, ink-lines on the floor [37], verbal feedback [33], active haptic feedback [16] and real-time visual feedback [12].” (lines 242-244).

Table 2: apologies but it is not clear what you mean by “I: 12, II: 7, III: 7, IV: 4”

This describes the number of people according to each KL grade; The table was revised as “KL grade, n (participants)” (Now, table 3) Please see table 3. (page 15).

Lines 273 to 274, “Heterogeneity values (I2) were estimated but not reliable (k=3)”: 

apologies, but I struggled to follow this. Heterogeneity is not a measure of reliability. Can I suggest you to re-word this to improve clarity and flow? There are other sections in your manuscript in which you reported heterogeneity in a similar way.

This has been revised to improve clarity in the methods section. This now reads “Statistical heterogeneity (I2) values were considered unreliable if less than seven papers were included in a meta-analysis [27]. “ (lines 189-190). 

This has been revised in the results section as “Statistical heterogeneity values (I2) were estimated but these values were not reliable as there were very few studies (k=3).” (line 299-301). 

Line 378, “We found with very low”: with – typo?

This has been revised now (line 408)

Line 406, “may have an impact”: I suggest you being more specific. This is a generic statement. Having an impact could be for the better or worse.

This has been revised. This now reads “ Although only reported in an observational study, participants who naturally walked with a toe-out gait pattern [43], had less medial knee OA structural disease progression over 18 months compared to those who walked with a more neutral FPA, suggesting that larger toe-out is associated with lower knee OA disease progression” (lines 433-438).

Line 415, “In clinical practice, consider that toe-in”: please revise sentence for flow and clarity

This has been revised. This now reads “When applying these gait interventions in clinical practice, it should be considered that toe-in reduces KAM1 but increases KAM2. “ (lines 446-448)

Line 434, “can reduce indicators of knee”: is it appropriate to state that? Your findings suggest very low to low certainty of evidence, with most interventions having a small effect. Taking these into account, I wonder whether your statement is too optimistic and supportive of those interventions. Later in that same paragraph, you adopt a more cautious interpretation (e.g. “While the certainty of the evidence is very low to low, there is potential for these interventions to be clinically helpful.”). my suggestion it to keep this cautious way of interpreting data throughout the manuscript.

This has been revised. This now reads “Although more work is needed to form clinical recommendations, gait strategies including ipsilateral trunk lean, toe-out and toe-in have a potential to reduce indicators of knee joint load in people with medial knee OA. Medial knee thrust, medial weight transfer and wider steps also may reduce knee load based on individual studies’ results.” (lines 479-483).

Review Comments to the Author

Reviewer #1: 

In some of the Tables there are some instances of NM rather than Nm. Please correct.

Thank you. This has been revised in Table 4.

Line 308-309: “A study which intervention lasting for 6-weeks…”- reword

Thank you. This has been revised. This now reads “A study with an intervention duration of 6-weeks…” (lines 336-337)

---

## [Editor Report · Decision Letter 1]

29 Jun 2022

PONE-D-21-38506R1Effects of neuromuscular gait modification strategies  on indicators of knee joint load in people with medial knee osteoarthritis: A systematic review and meta-analysisPLOS ONE

Dear Dr. Silva,

Thank you for submitting your manuscript to PLOS ONE. After careful consideration, we feel that it has merit but does not fully meet PLOS ONE’s publication criteria as it currently stands. Therefore, we invite you to submit a revised version of the manuscript that addresses the points raised during the review process.

We look forward to receiving your revised manuscript.

Kind regards,

Daniel Ribeiro

Academic Editor

PLOS ONE

Journal Requirements:

Additional Editor Comments (if provided):

Thank you for the revised version.

I have only few minor comments that I would ask you to revise.

"Statistical heterogeneity (I2) values were considered unreliable if less than seven papers were included in a meta-analysis [27].": My understanding is that it is not the I2 that is unreliable in this situation. The findings from your meta-analyses are unreliable if I2 is high. If you have small number of studies in a meta-analyses, then your I2 is likely to be biased (not unreliable). The terms "bias" and "reliable" reflect have different meanings.

In my previous comments, I suggested you to discuss the clinical heterogeneity of included studies in meta-analyses. At that time, I wrote "You could have discussed the likely impact of heterogeneity between included studies from a clinical and methodological point of view." I still believe this is important and you could discuss this and how or whether clinical heterogeneity is likely to have impacted in your findings. If clinical heterogeneity may have impacted on findings, how should the reader interpret it. This paper discusses this and may be of interest: https://bmcmedresmethodol.biomedcentral.com/track/pdf/10.1186/s12874-016-0121-7.pdf.
---

## [Author Response · Author response to Decision Letter 1]

10 Aug 2022

Rebuttal in response to reviewers

Revision- PONE-D-21-38506R1: Effects of neuromuscular gait modification strategies on indicators of knee joint load in people with medial knee osteoarthritis: A systematic review and meta-analysis

Author’s name: M Denika C Silva email: denika.silva@canberra.edu.au

We greatly appreciate your time in reviewing the manuscript and the detailed comments provided which are helpful to improve the paper.

Please see the following point-by-point responses to each comment raised by the academic editor.

Editor comments are in black font followed by the author's responses in blue font.

Please note the line numbers and page that we mention in this reply, refer to the manuscript with track changes.

Journal Requirements:

Author response: We reviewed the reference list and can confirm that it is complete and correct. The reference list does not include any papers that have been retracted.

Thank you very much to the editor, for the feedback and comments on the manuscript. The comments are very valuable and very much appreciated.

Editor Comments:

"Statistical heterogeneity (I2) values were considered unreliable if less than seven papers were included in a meta-analysis [27].": My understanding is that it is not the I2 that is unreliable in this situation. The findings from your meta-analyses are unreliable if I2 is high. If you have small number of studies in a meta-analyses, then your I2 is likely to be biased (not unreliable). The terms "bias" and "reliable" reflect have different meanings.

Author response:

Thank you, yes the terms bias and reliability were being used too loosely. This has been revised now, in the methods “Statistical heterogeneity (I2) values were estimated but may be biased if a smaller number of studies was included in meta-analyses [27].” (lines 170 to171)

In addition, this has been revised in the results. “Statistical heterogeneity values of I2=0, may be biased as there were very few studies included (k=3) [27].” (lines 279 to 281)

Editor comment:

In my previous comments, I suggested you to discuss the clinical heterogeneity of included studies in meta-analyses. At that time, I wrote "You could have discussed the likely impact of heterogeneity between included studies from a clinical and methodological point of view." I still believe this is important and you could discuss this and how or whether clinical heterogeneity is likely to have impacted in your findings. If clinical heterogeneity may have impacted on findings, how should the reader interpret it. This paper discusses this and may be of interest: https://bmcmedresmethodol.biomedcentral.com/track/pdf/10.1186/s12874-016-0121-7.pdf.

Author response:

Thank you for the feedback and for recommending a useful paper on clinical heterogeneity. 

In this review, we aimed to determine the effects of neuromuscular gait modification strategies on indicators of medial knee joint load in people with medial knee OA. Clinical variability characteristics include participants, the types or timing of outcome measurements, and the intervention (Cochrane handbook). All participants had medial knee OA, the severity of which, by study, is described in Table 3. We did not observe any effect due to KL grade, but the described ranges were wide. Other participant characteristics such as age and BMI varied, but not widely, since people with medial knee OA are a fairly homogenous population with older age and moderate to high BMI being a characteristic. Finally, although intervention timing was a primarily single session, there were longer-term interventions included. We did not analyse these separately because of the low number of studies. However, observation of the effect sizes did not reveal a marked differential effect. The intervention times are labelled on the forest plots and included in Table 4. Therefore, the reader can interpret the results of the meta-analyses in context. 

The impact of clinical heterogeneity has now been included in our limitations (lines 457 to 470). “Moreover, the results of the meta-analyses of this review potentially have been affected by clinical heterogeneity of included studies; including treatment duration, dose, and whether specific KAM feedback is used. The main source of clinical heterogeneity is probably intervention time. However, the majority of studies reported single-session data, and the forest plots did not suggest that intervention time significantly altered the effect size. Therefore, the clinical implication is that a single session is likely to be effective, however, the longer-term effects of the strategies are not yet well understood. The effects of trunk lean and toe-out seem to be enhanced by larger angles and the effect of toe-in is likely enhanced by specific KAM feedback. Therefore, when applying the findings of this review to practice, clinical heterogeneity in intervention suggests that trunk lean and toe-out with greater angles, and toe-in with specific KAM feedback may be more effective.”

---

## [Editor Report · Decision Letter 2]

7 Sep 2022

Effects of neuromuscular gait modification strategies  on indicators of knee joint load in people with medial knee osteoarthritis: A systematic review and meta-analysis

PONE-D-21-38506R2

Dear Dr. Silva,

We’re pleased to inform you that your manuscript has been judged scientifically suitable for publication and will be formally accepted for publication once it meets all outstanding technical requirements.

Kind regards,

Daniel Ribeiro

Academic Editor

PLOS ONE
---

## [Editor Report · Acceptance letter]

11 Sep 2022

PONE-D-21-38506R2 

Effects of neuromuscular gait modification strategies on indicators of knee joint load in people with medial knee osteoarthritis: A systematic review and meta-analysis 

Dear Dr. Silva:

I'm pleased to inform you that your manuscript has been deemed suitable for publication in PLOS ONE. Congratulations! Your manuscript is now with our production department. 

Kind regards, 

on behalf of

Dr. Daniel Ribeiro 

Academic Editor

PLOS ONE